# A Novel Antithrombocytopenia Agent, *Rhizoma cibotii*, Promotes Megakaryopoiesis and Thrombopoiesis through the PI3K/AKT, MEK/ERK, and JAK2/STAT3 Signaling Pathways

**DOI:** 10.3390/ijms232214060

**Published:** 2022-11-14

**Authors:** Wang Chen, Linjie Zhu, Long Wang, Jing Zeng, Min Wen, Xiyan Xu, LiLe Zou, Feihong Huang, Qianqian Huang, Dalian Qin, Qibing Mei, Jing Yang, Qiaozhi Wang, Jianming Wu

**Affiliations:** 1School of Pharmacy, Southwest Medical University, Luzhou 646000, China; 2School of Basic Medical Sciences, Southwest Medical University, Luzhou 646000, China; 3Education Ministry Key Laboratory of Medical Electrophysiology, Southwest Medical University, Luzhou 646000, China

**Keywords:** thrombocytopenia, *Cibotii rhizoma*, megakaryocyte differentiation, network pharmacology, PI3K/AKT, MEK/ERK, JAK2/STAT3

## Abstract

Background: *Cibotii rhizoma (CR)* is a famous traditional Chinese medicine (TCM) used to treat bleeding, rheumatism, lumbago, etc. However, its therapeutic effects and mechanism against thrombocytopenia are still unknown so far. In the study, we investigated the effects of aqueous extracts of *Cibotii rhizoma* (AECRs) against thrombocytopenia and its molecular mechanism.Methods: Giemsa staining, phalloidin staining, and flow cytometry were performed to measure the effect of AECRs on the megakaryocyte differentiation in K562 and Meg-01 cells. A radiation-induced thrombocytopenia mouse model was constructed to assess the therapeutic actions of AECRs on thrombocytopenia. Network pharmacology and experimental verification were carried out to clarify its mechanism against thrombocytopenia. Results: AECRs promoted megakaryocyte differentiation in K562 and Meg-01 cells and accelerated platelet recovery and megakaryopoiesis with no systemic toxicity in radiation-induced thrombocytopenia mice. The PI3K/AKT, MEK/ERK, and JAK2/STAT3 signaling pathways contributed to AECR-induced megakaryocyte differentiation. The suppression of the above signaling pathways by their inhibitors blocked AERC-induced megakaryocyte differentiation. Conclusions: AECRs can promote megakaryopoiesis and thrombopoiesis through activating PI3K/AKT, MEK/ERK, and JAK2/STAT3 signaling pathways, which has the potential to treat radiation-induced thrombocytopenia in the clinic.

## 1. Introduction

With the development of science and technology, ionizing radiation (IR) has been widely used in various fields of modern society, including nuclear power plants, nuclear weapons, X-ray medical diagnosis, tumor treatment, etc. However, while it brings benefits to people, it also causes adverse effects on human health [1]. In clinical practice, the most common IR injury is radiation therapy for malignant diseases such as tumors. Because during treatment, radiation kills tumor cells and damages normal tissue cells as well [2]. The hematopoietic system is vulnerable to damage in response to ionizing radiation [3]. Total body irradiation can damage the hematopoietic system, and eventually manifest as acute radiation syndrome (ARS) beyond a certain radiation dose. Some studies have reported that acute radiation doses of more than 1.7 Gy cause a decrease in the number of blood cells, increased infection and bleeding, and possibly death from severe neutropenia and thrombocytopenia [4]. At present, according to the U.S. Food and Drug Administration (FDA)-approved, Neupogen^®^, Neulasta^®^, Leukine^®^, and Nplate^®^ have been used to cure acute radiation syndrome (ARS) [5]. However, they are mainly used for the recovery of neutrophils [6]. Therefore, it is necessary to find drugs for the treatment of radiation-induced thrombocytopenia.

Platelets (PLTs) are anucleate cytoplasts that participate in hemostasis, which are produced by megakaryocytes. From megakaryocyte maturation to platelet production, megakaryocytes undergo multiple rounds of endomitosis, cytoplasmic expansion, demarcation membrane system (DMS) formation; an increase in cytoplasmic proteins and granules; and a massive cytoskeletal-driven reorganization in the megakaryocyte cytoplasm. Then, proplatelets are generated from mature megakaryocytes, which further mature into PLTs in sinusoidal blood vessels [7,8,9]. Meanwhile, this process is also regulated by a variety of cytokines, the most critical of which is thrombopoietin (TPO) [9]. TPO and its receptor c-mpl were discovered and cloned in 1994 and have been shown to promote the differentiation and maturation of hematopoietic stem cells (HSCs) into megakaryocytes [10,11,12,13,14,15,16,17]. Specifically, the combination of TPO and c-mpls receptor ligands initiates a variety of downstream signal transduction pathways, including JAK2, STAT3/STAT5, MAPK/ERK, and PI3K/AKT [18,19,20,21].

At present, the common treatment for thrombocytopenia is divided into non-drug therapy, such as platelet infusion, and drug therapy, such as thrombopoietin receptor ag-onists (TPO-RAs), recombinant human interleukin-11 (rHuIL-11), and recombinant human thrombopoietin (rHuTPO). Platelet transfusion can only increase platelets in the short term, but cannot maintain platelet stability in the long term, and there is a certain risk of blood disease infection or ineffective infusion [22]. Currently, TPO-RAs, romiplostim, eltrombopag, avatrombopag, and lusutrombopag have been approved by the FDA and European Medicines Agency (EMA) to increase platelet counts in a variety of conditions, such as immune thrombocytopenia (ITP) and severe aplastic anemia [23]. They represent a class of platelet growth factors that promote megakaryocyte growth, differentiation, and platelet production by mimicking the effects of endogenous TPO [24]. However, long-term use also can appear adverse reaction, such as thromboembolism, bone marrow reticular fibrosis, transaminase elevation, etc. [25,26]. The drug rhTPO has shown promise in early clinical trials [27,28], but it has developed cross-reactive antibodies that neutralize native human TPO in some subjects [29,30]. Thus, the development of the drug was halted in 2001. However, recently, the potential benefits of TPO have been explored again in China [31]. Recombinant human interleukin-11 (rhIL-11, oprelvekin) received FDA approval for the prevention of chemotherapy-induced thrombocytopenia (CIT) in patients with nonmyeloid malignancies [32]. However, it has significant toxicity and high cost [32]. Therefore, searching for new drugs to treat thrombocytopenia is necessary. Many Chinese medicinal materials (TCMs) are derived from natural products, so it is potentially possible to find new therapeutic drugs for thrombocytopenia from natural products. Many studies have underlined natural products for the treatment of thrombocytopenia because they can be used for a long time with low cost and fewer side effects compared with chemically synthesized drugs [33].

*Cibotii rhizoma (CR)* is the rhizome of *Cibotium barometz*, which is an evergreen fern belonging to the family Dicksoniaceae. It is a relatively practical and common traditional Chinese medicine (TCM) that has many pharmacological effects. From the perspective of TCM, it is often used to expel wind and dampness, reinforce benefits of the liver and kidneys, strengthen the waist and knees, prevent osteoporosis, treat traumatic bleeding, etc. In modern clinical applications, *CR* can treat waist, leg pain, hemiplegia, bleeding and osteoma, osteosarcoma, multiple myeloma, etc. [34]. Furthermore, the hairs of *CR* can be used as an anti-bleeding agent in wound-healing plasters [35]. It also has properties that nourish bones and improve gonadal function [35]. Although *CR* can treat blood diseases, including wound hemostasis and myeloma, the specific mechanism of action against thrombocytopenia is still unclear.

Here, we evaluate the effects of aqueous extracts of *Cibotii rhizoma* (AECRs) on megakaryopoiesis and thrombopoiesis, research the mechanism of action of AECRs against radiation-induced thrombocytopenia, and provide a theoretical foundation for its clinical application against thrombocytopenia.

## 2. Results

### 2.1. Characterization of Chemical Constituents in the Aqueous Extracts of Cibotii rhizoma

By using tandem high-resolution mass spectrometry and ultrahigh-performance liquid chromatography (UHPLCHRMS), the components of the aqueous extracts of *Cibotii rhizoma* (AECRs) were inferred. Figure 1A,B display the chromatograms of the total ion current for the AECRs. A total of 17 compounds were tentatively characterized in the constituents of AECRs, which are presented in Table 1. The structural formulas of these components are presented in Figure 1C.

### 2.2. Cytotoxicity of AECRs in K562 and Meg01 Cells

To ascertain whether AECRs exerted cytotoxicity on K562 and Meg01 cells, CCK-8, LDH, and flow cytometry assays were performed. After K562 and Meg01 cells were exposed to AECR (200, 300, and 400 μg/mL) for 4 days, we found that the cell viability of the high-concentration group (400 μg/mL) drastically decreased in comparison to the control group (Figure 2A). However, the high-concentration group (400 μg/mL) was safe or non-toxic. (Figure 2B). Meanwhile, on day 4, the concentration of AECRs (400 μg/mL) did not cause apoptosis compared to the control group (Figure 2C). Therefore, AECR concentrations of 400, 300, and 200 μg/mL were selected for subsequent tests.

### 2.3. AECRs Induce Megakaryocyte Differentiation and Maturation

In the process of differentiating and maturating megakaryocytes, the cell volume increases (polyploid cells), the cell nucleus is multinucleated and lobulated, the cells undergo endomitosis (resulting in polyploid nuclei), and the expression of cell surface antigens CD41 and CD42b increases. Thus, white light photography showed that many polyploid cells appeared in the PMA-positive (1.25 nM) group and the AECR (200, 300, and 400 μg/mL) treatment groups, but few big cells appeared in the control group in both K562 and Meg01 cells after treatment for 4 days (Figure 3A). Giemsa staining revealed that AECRs (200, 300, and 400 μg/mL) and PMA (1.25 nM) significantly increased the cell size and number of nuclei of K562 and Meg01 cells (Figure 3B). Phalloidin staining showed similar results: AECRs (200, 300, and 400 μg/mL) and PMA increased the size and multiple nuclei of K562 and Meg01 cells and upregulated F-actin (Figure 3C). Flow cytometry revealed that CD41^+^ CD42b^+^ increased in a dose-dependent manner in both K562 and Meg01 cells after 4 days of AECR treatment (Figure 3D). In addition, the ploidy assay demonstrated that the proportion of 2N cells in the AECR (200, 300, and 400 μg/mL) and PMA groups were significantly lower than in the control group; conversely, the proportion of 4N cells in the AECR (200, 300, and 400 μg/mL) and PMA groups was significantly higher than in the control group (Figure 3E), indicating that AECRs increased the ploidy of K562 and Meg01 cells. Taken together, these data demonstrate that AECRs can induce the megakaryocyte differentiation and maturation of K562 and Meg01 cells.

### 2.4. AECRs Accelerate PLT Recovery in Mice with Thrombocytopenia

To assess the activity of AECRs in vivo, a thrombocytopenia mouse model was created via total body irradiation with a 4 Gy X-ray (Figure 4A). After the mice were irradiated, the PLT level dropped to bottom on day 7 and gradually recovered to normal on day 14 (Figure 4B). The PLT levels of the TPO-positive group and AECR (143 mg/kg, 286 mg/kg, and 429 mg/kg) treatment groups exceeded those of the model group on days 10 and 12 (Figure 4B), suggesting that AECRs could promote PLT recovery in mice with thrombocytopenia. There was no obvious difference in the mean platelet volume (MPV) (Figure 4C) or red blood cells (RBCs) (Figure 4D), indicating that AECRs did not affect MPV or RBCs. The white blood cell (WBC) counts in the AECR-treated (143 mg/kg, 286 mg/kg, and 429 mg/kg) groups significantly exceed those in the model group on day 12, and the AECR-treated (429 mg/kg) group also exceeded the model group on day 14 (Figure 4E), suggesting that AECRs had some effects on WBC recovery. Meanwhile, the expression of the peripheral blood PLT markers CD41, CD61, and CD62P were analyzed on day 10. The proportions of CD41^+^ and CD61^+^ PLTs (Figure 4F) and CD41^+^ and CD62P^+^ PLTs (Figure 4G) in the TPO-positive group and AECR (143 mg/kg, 286 mg/kg, and 429 mg/kg) treatment groups were significantly increased compared with the model group, which indicates that AECRs accelerated PLT recovery in peripheral blood. Meanwhile, the safety evaluation of AECRs for organs is shown in Appendix A, which reveals that AECRs are safe and non-toxic. Altogether, these results suggest that AECRs stimulate PLT recovery in mice with thrombocytopenia and produce no systemic toxicity.

### 2.5. AECRs Promote Megakaryocyte Differentiation and Maturation in Mice with Thrombocytopenia

Megakaryocytes are mainly distributed in bone marrow (BM) and partly in the spleen, which produces and releases PLTs [36]. Therefore, megakaryocytes were detected in the BM and spleen. H&E staining of the BM (Figure 5A) and spleen (Figure 5B) on day 10 showed that megakaryocytes were significantly increased in the TPO-positive group and AECR (143 mg/kg, 286 mg/kg, and 429 mg/kg) treatment groups versus the model group. Immunohistochemical data further revealed that the proportions of CD41-positive megakaryocytes (Figure 5C) and VWF-positive megakaryocytes (Figure 5D) were increased after AECR (143 mg/kg, 286 mg/kg, and 429 mg/kg) and TPO treatment. These data suggest that AECRs induce megakaryopoiesis in the BM and spleen. Furthermore, the expression of the hematopoietic progenitor marker c-Kit and megakaryocyte marker CD41 was examined using flow cytometry. The results indicate an increase in the proportion of c-Kit^+^ CD41^+^ cells after AECR (143 mg/kg, 286 mg/kg, and 429 mg/kg) and TPO treatment (Figure 5E), demonstrating that AECRs stimulated the production of hematopoietic progenitors and megakaryocytic progenitors. AECR (143 mg/kg, 286 mg/kg, and 429 mg/kg) and TPO treatment obviously enhanced the proportion of CD41^+^ CD61^+^ cells (Figure 5F), which suggests that AECRs were able to induce megakaryocyte differentiation in BM. The ploidy assay revealed that AECR (143 mg/kg, 286 mg/kg, and 429 mg/kg) and TPO treatment obviously enhanced DNA ploidy 4N, 8N, and 16N in BM (Figure 5H) and also DNA ploidies 4N and 8N in the spleen (Figure 5I) compared with the model group. In addition, CD41 and CD62P expression in BM cells was analyzed, demonstrating that AECR (143 mg/kg, 286 mg/kg, and 429 mg/kg) and TPO treatment groups promoted the proportion of CD41^+^ CD62P^+^ cells (Figure 5G), indicating that AECRs can stimulate thrombopoiesis in BM. In conclusion, all these data demonstrate that AECRs can enhance megakaryocyte differentiation and maturation, as well as thrombopoiesis, in thrombocytopenia mice.

### 2.6. Network Pharmacology Prediction of the Targets and Mechanisms of AECRs against Thrombocytopenia

Seventeen active ingredients in the AECRs were identified (Table 1), and 383 targets of these ingredients were obtained (Appendix A). A total of 1300 targets related to thrombocytopenia were identified (scores > 1) (Appendix A). A total of 101 overlapping targets between AECRs and thrombocytopenia were acquired (Appendix A) and a Venn diagram is shown in Figure 6A.

Then, the networks of AECR anti-thrombocytopenia were constructed. As shown the Figure 6B, the AECR–ingredient–target–thrombocytopenia network revealed the relationships among drugs, targets, and ingredients. In this network, we found that some targets may be associated with multiple ingredients, and one ingredient may be associated with multiple targets. Similarly, in Figure 6C, 19 key targets were identified through the three parameters of degree (≥23), betweenness centrality (BC, ≥0.004028), and closeness centrality (CC, ≥0.443925) in the PPI network, including PIK3R1, SRC, JAK1, JAK2, TLR4, HSP90AA1, LYN, TNF, MAPK1, LCK, EGFR, CTNNB1, AKT1, PIK3CA, STAT3, PTPN11, MAPK8, RELA, and FYN. 

Next, the underlying mechanisms of AECRs against thrombocytopenia were predicted. The GO analyses result was show in the Figure 6D. Nineteen biological processes were screened out by setting the key parameters *p* < 0.01 in the Cytoscape plug-in ClueGo, including the positive regulation of cell adhesion (21.35%), the positive regulation of migration (13.48%), leukocyte differentiation (8.99%), phosphatidylinositol 3-kinase signaling (7.87%), PLT activation (4.49%), the regulation of MAP kinase activity (4.49%), the regulation of the G1/S transition of the mitotic cell cycle (1.12%), and the receptor signaling pathway via JAK-STAT (1.12%), which were all closely related to megakaryocyte differentiation and PLT production. The results from KEGG analyses are show in Figure 6E, and 11 enriched pathways were obtained through setting the key parameters *p* < 0.05, including PI3K/AKT, HIF-1, JAK/STAT, RAS, VEGF, MAPK, mTOR, etc., which were also all closely related to megakaryocyte differentiation and PLT production. 

Finally, this relationship between drugs, ingredients, diseases, and pathways is summarized in Figure 6F.

### 2.7. AECRs Induce Megakaryocyte Differentiation through the Activation of the PI3K/Akt, MAPK, and JAK/STAT Signaling Pathways

The results of network pharmacology analysis were proven by Western blot. The expression of key target proteins was validated. According to these results, we found that AECRs could significantly up-regulate the expression of proteins (IRS1, PI3K, AKT, and mTOR) related to the PI3K/AKT signaling pathway (Figure 7A), proteins (RAS, MEK, and ERK) related to the MAPK signaling pathway (Figure 7B), proteins (JAK2, STAT3) related to the JAK/STAT signaling pathway (Figure 7C), and several hematopoietic transcription factors (TAL1, PBX1, and MEIS1) and inhibited the expression of c-Myb (Figure 7D). To further verify the underlying multitarget and multi-pathway effects of AECRs, the PI3K/AKT signaling pathway inhibitor LY294002, the JAK/STAT signaling pathway inhibitor ruxolitinib phosphate, and the MEK-ERK signaling pathway inhibitor SCH772984 were used. As expected, treatment with LY294002 (Figure 8A), ruxolitinib phosphate (Figure 8B), and SCH772984 (Figure 8C) markedly blocked the content of CD41 and CD42b induced by AECRs. To sum up, these data demonstrate that AECRs promote megakaryocyte differentiation through the activation of the PI3K/AKT, MAPK, and JAK/STAT signaling pathways.

## 3. Discussion

Total body ionizing radiation may result in acute injury on hematopoiesis, including severe neutropenia and thrombocytopenia, which can be life-threatening [37]. BM is a key to the hematopoietic system and a vital organ for the production of PLTs. Currently, several treatments of PLTs have focused on attenuating the damage inflicted upon the BM or promoting megakaryocyte differentiation, which accelerates the production of PLTs. In this study, our data show that AECRs promoted megakaryocyte differentiation in K562 and Meg-01 cells in vitro and that AECRs were able to accelerate platelet recovery, megakaryopoiesis, and thrombopoiesis in vivo. We concluded that AECRs ameliorate radiation-induced thrombocytopenia by promoting megakaryocyte differentiation and platelet production, as shown in Figure 9.

As a TCM, CR contains many pharmacological ingredients. In the previous study, it contained protocatechuic acid, protocatechuic aldehyde, onitin, caffeic acid, resveratrol, kaempferol, formononetin, tannins, saponins, etc. In this study, our data show 17 potentially active ingredients in the AECRs by using UPLC-MS/MSUPLC-MS/MS. These compounds are monotropein, 4-methylcatechol, fraxetin, arbutin, epigallocatechin, aucubin, sinapic acid, caffeic acid, agnuside, esculetin, eriodictyol, dictamnine, 4-methyldaphnetin, herniarin, chrysophanol, fisetin, and maslinic acid. Among them, caffeic acid, eriodictyol, fisetin, and maslinic acid (MA) can treat blood diseases. Caffeic acid is effective in treating primary immune thrombocytopenia (ITP) and potentially against severe fever with thrombocytopenia syndrome (SFTS) virus (SFTSV) [38,39]. Epigallocatechin gallate in green tea has anti-multiple myeloma (MM) properties, and its metabolizing polyphenol compound eriodictyol can enhance the induction of epigallocatechin gallate apoptosis in vivo [40]. Fisetin is a class of flavonoids and is a common component of the human diet. It has been reported that a variety of cancers, such as chronic myeloid leukemia [41], promyelocytic leukemia [42], and multiple myeloma [43], can all be treated with fisetin. MA is a pentacyclic triterpene acid with many biological activities, such as anticancer, anti-inflammatory, and antioxidant activities. MA has been reported to have significant inhibitory activity on human leukemia cells (CCRF-CEM) and their multidrug-resistant sublines (CEM/ADR5000) [44].

Megakaryocyte differentiation is associated with a variety of mechanisms. In this study, PI3K/AKT/mTOR, RAS/MEK/ERK, and JAK-STAT activation were associated with megakaryocyte differentiation and platelet production. According to the report, they are all associated with hematopoiesis. Mitogen-activated protein kinases (MAPKs) are involved in regulating cellular responses to proliferation, migration, differentiation, and apoptosis [45]. Rapamycin is an inhibitor of mTOR that can inhibit megakaryocyte proliferation and differentiation. ERK1/2 belongs to the extracellular signal regulation of the MAPK family, which is essential for cell proliferation and differentiation. The activation of ERK1/2 is regulated by upstream mitogen-activated protein kinases (MAPKKs or MEKs). Many studies have reported that ERK1/2 is important to megakaryocyte proliferation, differentiation, and platelet formation [46,47,48]. The JAK2/STAT pathway is important in megakaryocyte proliferation and differentiation and is activated by cytokines such as TPO. When TPO binds to the receptor c-MPL, JAK2 is activated to regulate downstream signaling pathways, including the STAT, RAS/RAF-1/MAPK, and PI3K/AKT pathways [49,50,51,52,53]. Other cytokines, chemokines, and extracellular matrix proteins also affect megakaryopoiesis [54,55]. The transcription factors PBX1, MEIS1, TAL1, and c-Myb also play important roles in hematopoietic and megakaryocyte differentiation. PBX1 is a cofactor of BM echovirus integration site 1 (MEIS1) homologs and hematopoietic Hox transcription factors such as HoxA9. It has been reported that the deletion of PBX1 in mouse embryos leads to a decrease in the number of HSCs, BM progenitor cells, and megakaryocyte–erythrocyte progenitor cells [56,57]. MEIS1 is the target of the platelet-specific gene Pf4/Cxcl4, which promotes the development of megakaryocyte progenitors and inhibits the differentiation in red progenitor cells [58]. Meanwhile, MEIS1 transcription is directly regulated by growth factor independent 1B (Gfi1b), a zinc finger transcription factor that is critical to the development of erythrocyte and megakaryocyte lineages [59,60]. The conditional knockout of TAL1 in the hematopoietic system results in the reduced specificity of erythrocytes and megakaryocytes, and the knockdown of TAL1 in megakaryocytes results in megakaryocyte proliferation inhibition, polyploid formation disorders, cytoplasmic maturation defects, and a reduced platelet count [61,62]. c-Myb plays a role in the proliferation of hematopoietic cells and the differentiation in hematopoietic lineages. c-Myb expression is increased in hematopoietic progenitor cells and decreased during differentiation [63]. Homozygous c-Myb mutant mice exhibit hematopoietic failure and embryo death at 15 days [64]. The antisense knockdown of c-Myb in human myeloid cells reduces colony size and the number of monocytes [65]. The c-Myb gene knockout of mice also showed that c-Myb expression is essential for T- and B-cell development, myogenesis, and erythropoiesis [66]. In this study, the target expression was verified by Western blot, and these signaling pathways were activated by AECRs. Meanwhile, K562 cells were treated with PI3K, ERK1/2, and JAK inhibitors, and the expression of CD41-CD42b, a specific marker of megakaryocyte differentiation and maturation, was decreased by flow cytometry. Based on the above studies, we can conclude that AECRs can treat thrombocytopenia by acting on multiple pathways and multiple targets to promote megakaryocyte differentiation and platelet generation.

In summary, although the current data suggest that AECRs may treat radiation-induced thrombocytopenia by promoting megakaryocyte differentiation and maturation, the argument for this conclusion is still worthy of consideration and further confirmation. In the study, although K562 (human chronic myeloid leukemia cell) and Meg01 (human megakaryocyte) cells are considered classical models for studying megakaryocyte differentiation, they are also leukemia cells and differ to some degree from human CD34 hematopoietic progenitor cells that produce PLTs. Human BM CD34 has been reported to produce megakaryocytes and platelets in vitro [67,68]. Therefore, the demonstration of the mechanism in vitro still requires a large number of biological replicates, as well as the construction of other cell models [69,70,71,72], such as human BM CD34 hematopoietic stem cells, for future demonstrations and studies. In vivo, our data suggest that AECRs can treat radiation-induced thrombocytopenia by promoting hematopoietic stem progenitor cells to mature into megakaryocytes. This conclusion is significant because TPO-RA drugs, such as romiplostim, eltrombopag, avatrombopag, and lusutrombopag, are currently approved by the FDA and EMA for the treatment of thrombocytopenia [23], and their mechanism is to promote megakaryocyte growth, differentiation, and platelet generation [24]. However, there are some limitations in the conclusions of this paper, such as the small sample size (n = 3) and shallow mechanism studies. Therefore, to further verify and deepen the mechanism by which AECRs promote megakaryocyte differentiation to produce platelets, a large number of biological repeats and other methods [73] are needed in the future to delve into the mechanism research.

## 4. Materials and Methods

### 4.1. The Preparation of Aqueous Extracts of Cibotii rhizoma

AECRs were gifted from Sichuan New Green Pharmaceutical Science and Technology Development Co., Ltd. (Chengdu, China, batch number: 21020060) and dissolved in sterile deionized water or normal saline (NS) to interfere with the cells or animals.

### 4.2. The Main Ingredients of Aqueous Extracts of Cibotii rhizoma

The ingredients of aqueous extracts of CR (AECRs) were characterized by UPLC-HRMS. The extract was dissolved in aqueous methanol (4:1, *v*:*v*) and centrifuged at 13000 g at 4 °C for 15 min, and then 2 μL of the supernatant was injected to the UHPLC-Q Exactive HRMS system. The HSS T3 column (100 mm × 2.1 mm i.d., 1.8 μm) was used, as well as 0.1% formic acid in acetonitrile, water, and isopropanol at a ratio of 95:5 (solvent A) and 47.5:47.5:5 (solvent B), respectively. The program for gradient elution was as follows: 0 to 2 min, 5% (B) → 25% (B); 2 to 9 min, 25% (B) → 100% (B); 9 to 12 min, 100% (B) to 0% (B). The flow rate was 0.4 mL/min. The column temperature was 40 °C. During the entire process, all these samples were stored at 4 °C. These ingredients were manually characterized based on the accurate mass, fragmentation patterns, and isotope ratio.

### 4.3. Cell Culture

K562 and Meg01 cells were purchased from the ATCC cell bank (Bethesda, MD, USA) and cultivated in a complete RPMI 1640 medium (Gibco, Invitrogen Corporation, Carlsbad, CA, USA) with 10% FBS and 1% penicillin–streptomycin (Beyotime, Sichuan, China). At 37 °C, the cells were kept in a 5% CO_2_-humidified environment during incubation.

### 4.4. Cell Proliferation Assay

The CCK-8 assay was used to evaluate how AECRs impacted the viability of K562 and Meg01 cells. First, a 96-well plate containing 3 × 10^3^ cells in each well was seeded with various AECR concentrations for 6 days (0, 200, 300, and 400 μg/mL) and the entire volume was 200 μL in each well. Then, each well received 20 μL of CCK-8 solution (Dojindo, Kumamoto, Japan), which was supplied to and incubated for the appropriate duration at 37 °C. The absorbance value was measured with an Epoch Multi-Volume Spectrophotometer System (BioTek, Winooski, VT, USA). The wavelength was λ = 450 nm.

### 4.5. Lactate Dehydrogenase (LDH) Assay

A 96-well plate containing 3 × 10^3^ cells in each well was seeded with various AECR concentrations for 6 days (0, 200, 300, and 400 μg/mL) and the entire volume was 200 μL in each well. Then, the cytotoxicity was evaluated on days 2, 4, and 6 by an LDH cytotoxicity assay kit (Beyotime, Jiangsu, China).

### 4.6. Cell Apoptosis Assay

The cells (3 × 10^4^ cells/mL) were incubated with AECRs (200, 300, and 400 µg/mL) for 4 days and then detected with an apoptosis kit (Vazyme, A211-01). In brief, cells were harvested for 5 min at 1200 rpm and resuspended in a 1 × Annexin V binding buffer working solution. Then, the cell suspension was stained with 5 µL of Annexin V and PI at room temperature without light for 30 min, respectively. The apoptosis rate was measured using a flow cytometer (BD Biosciences, San Jose, CA, USA).

### 4.7. Morphologic Analysis

For 4 days, the cells were incubated with AECRs (200, 300, and 400 μg/mL). Light microscopy was used to examine the cell morphology (Nikon, Tokyo, Japan).

### 4.8. Giemsa Staining

Giemsa staining was performed to observe the nuclear morphology. The cells were harvested at 1200 rpm for 5 min, and then 1 ml of KCL (0.075 mol/L) was added to swell them for 2–3 min. After being washed with cold phosphate-buffered saline (PBS), the cells were fixed in a mixture of methanol and glacial acetic acid solution (3:1, *v*:*v*) for 5 min. Then, it was centrifuged at 1200 rpm for 2 min, and 200 µL of the supernatant was left and gently mixed. Next, the cells were coated on glass slides, dried, stained for 8 min with Giemsa working solution in a mixture of Giemsa stock solution and Giemsa buffer = 1:9 (G1010), and then washed for 30 s. Finally, the sample was analyzed by light microscopy (Nikon, Japan).

### 4.9. Phalloidin Staining

The cells were planted on 6-well plates at 3 × 10^4^ cells/mL for 4 days with or without AECRs (200, 300, and 400 µg/mL). Following three PBS washes, the cells were fixed for 12 min with 4% paraformaldehyde. Then, following three washes with PBS, the cells were coated on glass slides, dried, permeabilized for five minutes with 0.1% Triton X-100, and then slowly washed with PBS for 3–5 times. Next, the cells were treated with 200 μL of TRITC phalloidin fluid (CA1610) for 1 h at room temperature without light. Cell nuclei were stained for five minutes at room temperature with DAPI (C0065) after three rounds of washing, slowly. Finally, images were immediately collected under an inverted fluorescence microscope (Nikon Ts2R/FL, Japan).

### 4.10. Measurement of Megakaryocyte Differentiation by Flow Cytometry

The cells (3 × 10^4^ cells/mL) were treated with or without AECRs (200, 300, and 400 µg/mL) for 4 days. The cells were then harvested and resuspended in 100 μL of ice-cold PBS, followed by the addition of 3 μL of FITC-CD41 and PE-CD42b antibodies (Biolegend, San Diego, CA, USA), respectively, and incubated for 30 min at room temperature in the dark. Thereafter, 1 × 10^5^ cells were resuspended in 500 μL of ice-cold PBS for flow cytometry analysis (BD Biosciences, San Jose, CA, USA). The cells (4 × 10^4^ cells/well) were treated with AECRs (400 µg/mL), LY294002 (20 μM), ruxolitinib phosphate (100 nM), or SCH772984 (2 μM) for 4 days. In brief, we set the following different groups for the experiment, including the control group, the inhibitor group, the AECR+ inhibitor group, and the AECR group. On the fourth day, the cells were treated and analyzed as described above.

### 4.11. Megakaryocyte Ploidy Analysis

Cells (1 × 10^5^) were harvested and resuspended in 1 ml of 70% ethanol for an overnight period at 4 °C in order to analyze the DNA ploidy. The cells were stained the following day using 400 µL of Pi/RNase staining buffer from BD Pharmingen in California, USA, for 30 min at 4 °C before being analyzed using flow cytometry (BD Biosciences, San Jose, CA, USA).

### 4.12. Animals

Specific pathogen-free (SPF) Kunming (KM) mice (18–22 g, 4 weeks old) were purchased from Ten-Xin Co. Ltd. (Chongqing, China) and kept in an SPF facility for animals. The room temperature was maintained at 25 ± 2 °C and there was a 12 h light–dark cycle. The mice received standard rodent food (Sichuan Chengdu Dashuo Biotechnology Co., Ltd., Chengdu, China). The Committee on Animal Use and Care of the Southwest Medical University approved all experimental operations involving animals (Luzhou, China).

### 4.13. Establishment of a Thrombocytopenia Mouse Model and Treatment with AECRs

According to the principle of random allocation, Kunming mice were divided into 6 groups, with 8 mice in each group, including the control and model group, the positive group, and three AECR groups (low dose, medium dose, and high dose, respectively), namely the normal + NS group, the IR + NS group, the IR+ TPO (3000 U/kg) group, and the IR+ AECR group (143 mg/kg, 286 mg/kg, and 429 mg/kg, respectively). These mice were subjected to a radiation-induced thrombocytopenia model, excluding a control group. Then, the six groups were given the following intervention: the control group and model group were intragastrically administered with normal saline, and the positive group of TPO was intraperitoneally administered TPO for 14 consecutive days. Different AECR concentrations were intragastrically administered to the other AECR groups for 14 consecutive days.

### 4.14. Hemanalysis

The day before medication was set as day 0. On days 0, 4, 7, 10, 12, and 14, blood samples from 8 mice per group were collected from the fundus vein plexus. Hematologic parameters were recorded by using a Sysmex XT-1800i automatic hematology analyzer (Kobe, Japan).

### 4.15. Histopathological Analysis

All mice were killed on day 10, and their femurs and spleens were taken out and fixed for a week in 4% paraformaldehyde. Subsequently, the samples were cut into slices with a microtome (RM2016, Leica, Shanghai, China), immersed in paraffin, and processed. The H&E stain was applied to the paraffin sections after being dewaxed and hydrated. Finally, pictures were captured using an Olympus BX51 microscope (Olympus Optical, Tokyo, Japan).

### 4.16. Immunohistochemical Staining of CD41 and VWF

After deparaffinization and rehydration, the femurs were immersed in citrate buffer (pH = 6.0). This was followed by intervention with 3% H_2_O_2_ and washing with PBS for 5 min three times. Then, the femurs were blocked with normal goat serum (3%) at 37 °C for 30 min and treated with the primary rabbit anti-rat antibody CD41 (1:100, Proteintech, Wuhan, China), VWF (1:100, Proteintech, Wuhan, China) overnight at 4 °C. Afterwards, they were treated with an enzyme-labeled goat anti-rabbit IgG secondary antibody for 60 min and stained with diaminobenzidine (DAB) (Servicebio, Wuhan, China); the sections were mounted, dehydrated, cleared, and counterstained with hematoxylin for 3 min. Finally, images were observed and photographed under an Olympus BX51 microscope (Olympus Optical).

### 4.17. Flow Cytometry Analysis of Bone Marrow (BM), Spleen Cells, and Peripheral Blood

Mouse BM cells were rinsed twice from the femur with 1 mL of saline and then filtered through nylon mesh (YA09600), and 125 μL of flow preservation solution was added. A small portion of splenic tissue was cut and ground on a nylon mesh and rinsed with 1 mL of normal saline. Then, the liquid was filtered again through a nylon mesh and 125 μL of flow preservation solution was added. Blood samples (50 μL) were suspended in the citric acid buffer (1 mL). All samples were stored in the refrigerator at 4 °C. Subsequently, 200 μL of liquid containing 1 million cells was taken from each of the samples, and the samples were labeled with 1.25 μL of FITC-CD41 (BioLegend, San Diego, CA, USA), 0.625 μL of PE- c-Kit (BioLegend, San Diego, CA, USA), 0.625 μL of PE- CD61, and 0.625 μL of CD62P (BioLegend, San Diego, CA, USA), and incubated at room temperature and in the dark for 15 min. Finally, a flow cytometer (BD Biosciences, San Jose, CA, USA) was used to analyze the samples.

### 4.18. Network Pharmacology

The active ingredients of AECRs and targets were collected. The active compounds were obtained by UPLC-MS/MS. Then, the active ingredient was entered into the PubChem database to obtain its SDF format and SMILES format, respectively. Next, the SDF format or SMILES format was imported into the Swiss Target Prediction database to obtain the targets of the ingredient, respectively. Finally, the non-repetitive targets of all components were obtained.

The intersection targets of thrombocytopenia and AECRs were screened. In the GeneCards database, thrombocytopenia targets were identified with the keywords “thrombocytopenia” or “thrombopenia”. Then, thrombocytopenia targets with a relevance score greater than 1 were screened. Next, we imported the target of the disease and the target of all components into the Venn diagram platform (https://bioinfogp.cnb.csic.es/tools/venny/) to obtain the common target on 15 July 2022. Finally, the “AECR—ngredients—target—thrombocytopenia” network was constructed by Cytoscape 3.9.1 (http://cytoscape.org/) on 15 July 2022.

A protein–protein interaction network (PPI) was constructed and the core targets were acquired. First, we imported the common target genes of drugs and diseases into the String 11.5 database (https://string-db.org) on 15 July 2022, and then set the biological species “Homo sapiens” and “confidence > 0.7” to obtain the protein–protein interaction file. Next, the PPI was visualized in Cytoscape 3.9.1. Finally, the core targets were obtained by analyzing three parameters, degree, BP, and CC, in the Cytoscape 3.9.1.

Gene ontology (GO) enrichment analysis was then conducted. The common targets of drugs and components were introduced into the ClueGo plug-in of Cytoscape 3.9.1, and then the main parameter *p* < 0.01 was set to obtain the GO enrichment map.

Kyoto Encyclopedia of Genes and Genomes (KEGG) enrichment analysis came next. The common targets of drugs and components were introduced into the DAVID database (https://david.ncifcrf.gov) to obtain all pathways on 15 July 2022. Then, we screened out the core pathways through *p* < 0.05. Finally, the core pathways were shown by a bubble plot in bioinformatics (www.bioinformatics.com.cn) on 15 July 2022.

### 4.19. Western Blot

The K562 cells were exposed to AECRs for 4 days, washed with PBS, and lysed in 1× RIPA lysis buffer (Cell Signaling Technology, Beverly, MA, USA) with an EDTA-free protease inhibitor cocktail (TargetMol, Shanghai, China). The protein concentrations were determined using a Bradford assay (Cat #5000205) after total proteins were collected from the cells. The solubilized cell lysates were transferred to polyvinylidene difluoride (PVDF) membranes (Millipore, Darmstadt, Germany) after being separated by SDS-PAGE and solubilized in an SDS sample buffer. Next, the membranes were immersed in 5% skim milk powder solution in PBS-Tween 20 (PBST) with gentle shaking for an hour and a half. The membranes were washed in PBST three times and incubated overnight at 4 °C separately with the indicated primary antibodies. Subsequently, these membranes were treated with a secondary antibody coupled to horseradish peroxidase (HRP), detected with UltraSignalTM ECL Western blotting detection reagents (Biotech Co., Ltd., Beijing, China), and captured using a ChemiDoc MP Imaging System (Bio-Rad, Hercules, CA, USA). The quantitative analysis of these bands (National Institutes of Health, Bethesda, MD, USA) was performed with the ImageJ program. Three duplicates of each experiment were run.

The primary antibodies were as follows: TAL1 (CST, 12831S), phospho (P)- MEK1/2 (CST, 2338S)), MEK1/2 (CST, 4694S), P-ERK1/2 (CST, 9101S), ERK1/2 (CST, 9194S), P-STAT3 (CST, 9145S), STAT3 (CST, 12640S), P-PI3K (CST, 17366S), PI3K (CST, 4255S), P-AKT (CST, 4060S), AKT (CST, 4685S), mTOR (Proteintech, 66888-1-Ig), RAS (CST, 67648S), ESR1 (Abmart, M011463S), IRS1 (Abmart, TA6273S), MEIS1 (Affinity Biosciences, DF8368) c-Myb (Proteintech, 17800-1-AP), PBX1 (Proteintech, 17800-1-AP), and β-actin (CST, 4970S). The ratio of antibodies that worked best was 1:1000.

### 4.20. Statistical Analysis

The data, which are displayed as the mean ± standard deviations, were examined using GraphPad Prism 9.0. Using one-way ANOVA, the differences between three or more groups were evaluated. When the differences reached *p* < 0.05, they were deemed statistically significant.

## 5. Conclusions

In this study, we obtained 17 possible components in AECRs for the first time through UHPLCHRMS. Second, network pharmacology predicted that the molecular mechanism of AECRs in the treatment of thrombocytopenia was closely related to the PI3K/AKT, JAK/STAT, RAS, MAPK, and mTOR pathways. In vitro experiments confirmed that AECRs could promote megakaryocyte differentiation by activating the PI3K/AKT, MEK/ERK, and JAK2/STAT3 signaling pathways. In vivo experiments confirmed that AECRs promote the differentiation and maturation of megakaryocytes, as well as platelet production. Finally, it can be concluded concluded that AECRs may treat radiation-induced thrombocytopenia by promoting megakaryocyte differentiation to produce platelets, thus indicating that it may be a new drug for the treatment of thrombocytopenia and lay a foundation for future clinical applications.

## Figures and Tables

**Figure 1 ijms-23-14060-f001:**
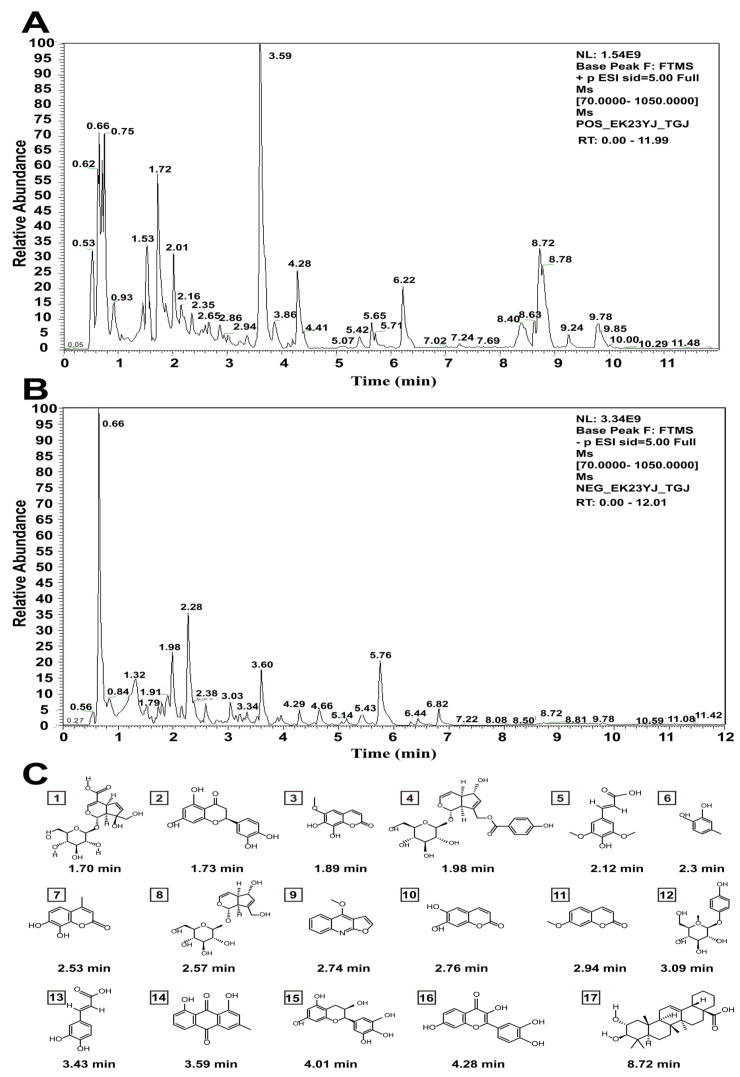
(**A**,**B**) The total ion current chromatogram of aqueous extracts of *Cibotii rhizoma* (AECRs): the positive mode (**A**); the negative mode (**B**). (**C**) Chemical structure of 17 compounds in AECRs.

**Figure 2 ijms-23-14060-f002:**
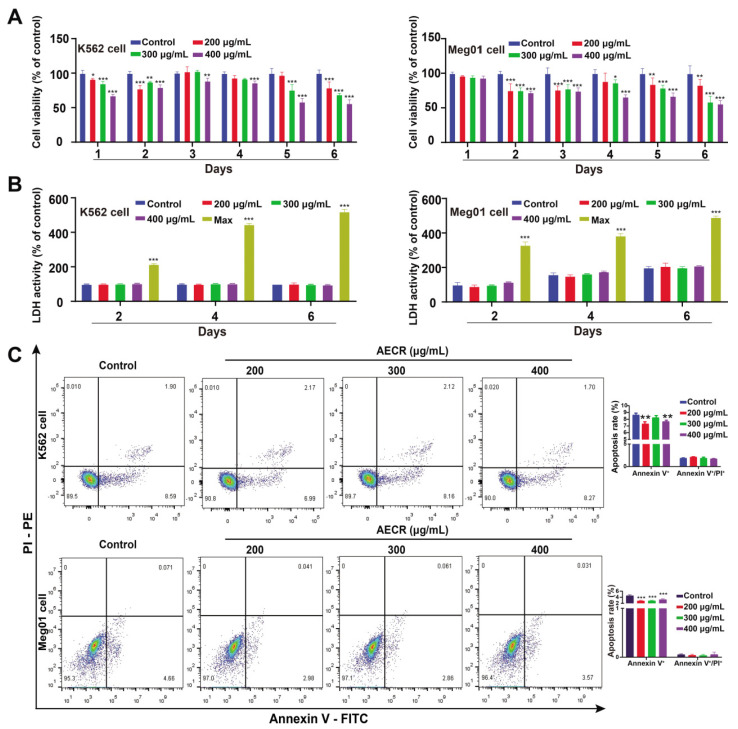
Determination of K562 and Meg01 cell concentrations. (**A**) Measurement of cell proliferation of K562 and Meg01 cells with the CCK-8 assay after AECR intervention. The values represent the means ± standard deviations (n = 3). (**B**) LDH assay for K562 and Meg01 cell cytotoxicity. Groups were set as follows: the control group, the AECR (200, 300, 400 μg/mL) group, and the max group (sample maximum activity control group). Additionally, 2, 4, and 6 represent the number of consecutive days of AECR intervention. The values represent the means ± standard deviations (n = 3). (**C**) After Annexin V-FITC/PI staining for K562 and Meg01 cells, the apoptosis was detected. The values represent the means ± standard deviations (n = 3). * *p* < 0.05, ** *p* < 0.01, *** *p* < 0.001 versus the control group. AECRs: aqueous extracts of *Cibotii rhizoma*; LDH: lactate dehydrogenase.

**Figure 3 ijms-23-14060-f003:**
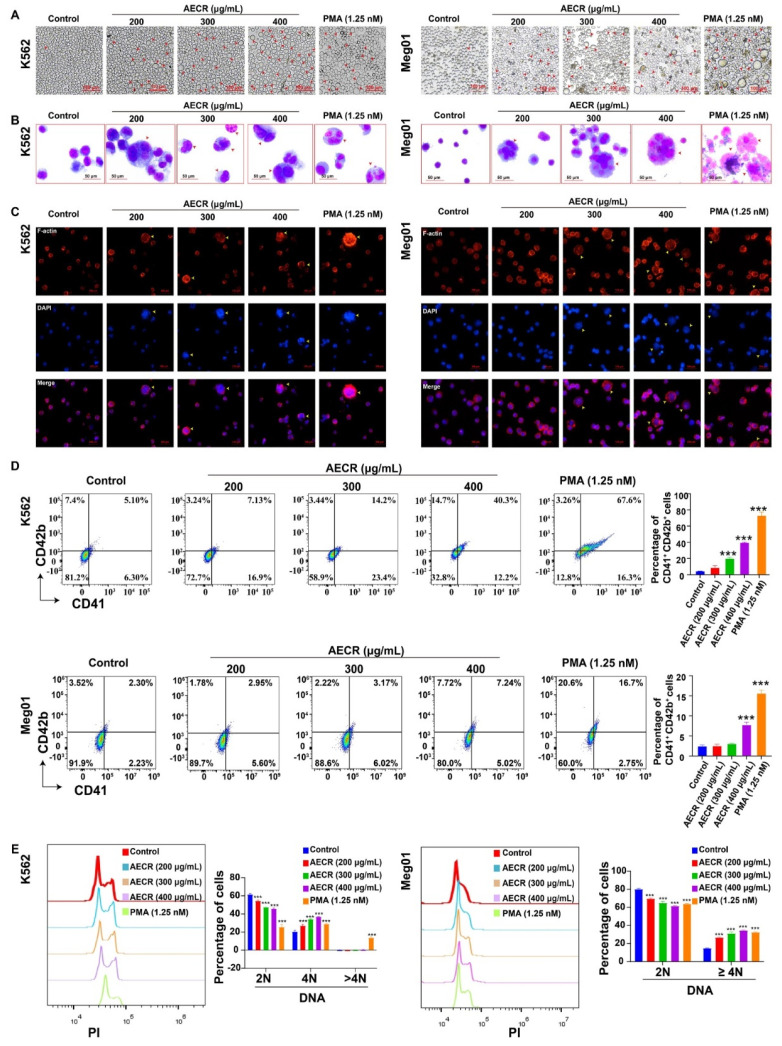
AECRs induce K562 and Meg01 cell differentiation and maturation. (**A**) Representative images show that cell size is increased after 4 days of AECR and PMA intervention (red arrowheads). (**B**) Cells stained with Giemsa showed large and numerous nuclei upon PMA or AECR treatment for 4 days (red arrowheads). (**C**) Cells stained with phalloidin showed the enhanced expression of F-actin upon PMA or AECR treatment for 4 days. (**D**) The CD41 and CD42b expression levels were analyzed by flow cytometry after 4 days of treatment with AECRs and PMA. On the right, this statistical histogram displays CD41^+^ CD42b^+^ cell proportions. The values represent the means ± standard deviations (n = 3). (**E**) After the cells were treated with AECRs for 4 days, the cells with PI staining were used to analyze DNA content (2, 4, and >4N) by flow cytometry. On the right, this statistical histogram displays cell proportions. The values represent the means ± standard deviations (n = 3). *** *p* < 0.001 versus the control group. AECRs: aqueous extracts of *Cibotii rhizoma*; PMA: phorbol-12-myristate-13-acetate.

**Figure 4 ijms-23-14060-f004:**
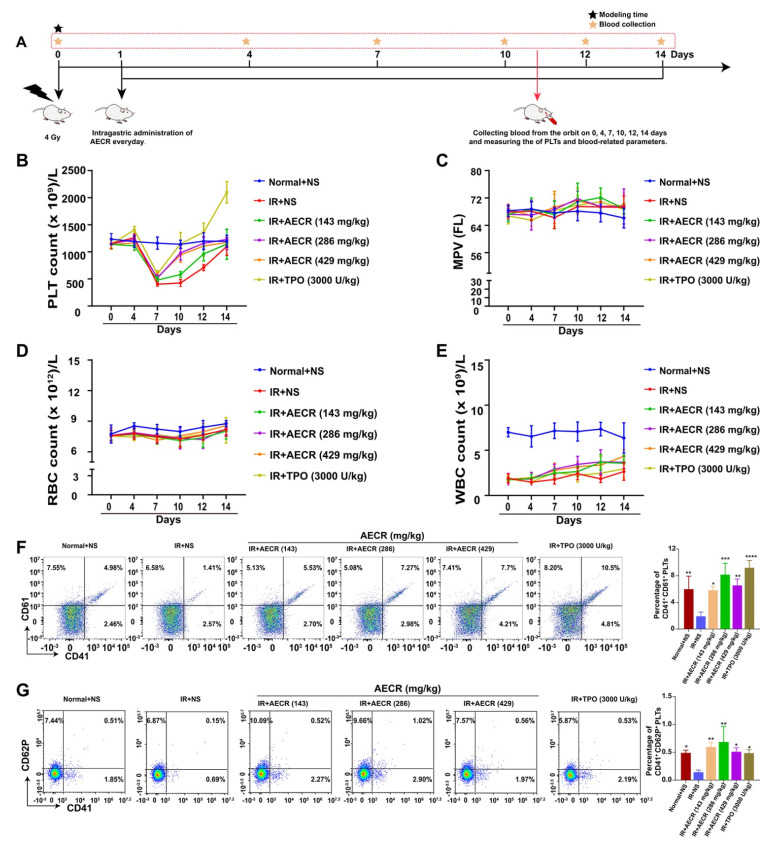
AECRs accelerate the recovery of peripheral blood PLTs in mice with thrombocytopenia. (**A**) Schematic diagram of animal experiment. (**B**–**E**) AECRs affect the PLTs, MPV, RBCs, and WBCs in peripheral blood. The values represent the means ± standard deviations (n = 8). * *p* < 0.05, ** *p* < 0.01, *** *p* < 0.001 versus the model group. (**F**,**G**) The CD41^+^ CD61^+^ and CD41^+^ CD62P^+^ content on platelets were analyzed by flow cytometry on day 10 after treatment with AECRs. On the right, these statistical histograms show the proportion of CD41^+^ CD61^+^ and CD41^+^ CD62P^+^ PLTs in each group. The values represent the means ± standard deviations (n = 3). * *p* < 0.05, ** *p* < 0.01, *** *p* < 0.001, **** *p* < 0.0001 versus model group. AECRs: aqueous extracts of Cibotii Rhizoma; PLTs: platelets; MPV: mean platelet volume; WBCs: white blood cells; RBCs: red blood cells.

**Figure 5 ijms-23-14060-f005:**
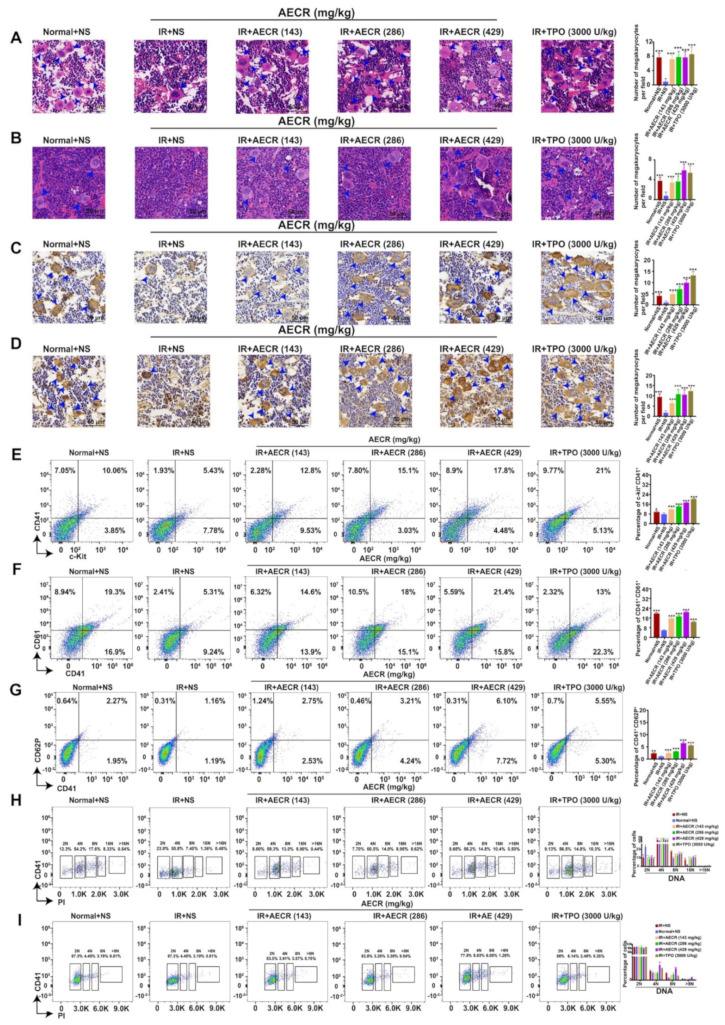
AECRs promoted the differentiation and maturation of BM and splenic megakaryocytes at 10 days. (**A**,**B**) H&E staining shows that the megakaryocytes from BM and the spleen were increased after AECR (143 mg/kg, 286 mg/kg, and 429 mg/kg) and TPO treatment (blue arrowheads). On the right, this statistical histogram shows cell proportions in each group. The values represent the means ± standard deviations (n = 3). (**C**,**D**) Immunohistochemical staining shows that the number of CD41-positive (Figure 5C) and VWF-positive (Figure 5D) megakaryocytes are increased after AECR (143 mg/kg, 286 mg/kg, and 429 mg/kg) and TPO treatment (blue arrowheads). On the right, this statistical histogram shows cell proportions in each group. The values represent the means ± standard deviations (n = 3). (**E**–**G**) Flow cytometry analysis of c-Kit^+^ CD41^+^, CD41^+^ CD61^+^, and CD41^+^ CD62P^+^ expressions on BM treated with AECRs (143 mg/kg, 286 mg/kg, and 429 mg/kg) and TPO. On the right, these statistical histograms show the ratio of c-Kit^+^ CD41^+^, CD41^+^ CD61^+^, and CD41^+^ CD62P^+^ cells in each group. The values represent the means ± standard deviations (n = 3). (**H**) BM cells were stained with CD41^+^ and PI, and DNA content (2, 4, 8, 16, and >16N) was determined by flow cytometry. On the right, this statistical histogram shows the ratio of cells in each group. The values represent the means ± standard deviations (n = 3). (**I**) The spleen cells were stained with CD41^+^ and PI, and flow cytometry was used to calculate DNA content (2, 4, 8, and >8 N). On the right, this statistical histogram shows cell proportions in each group. The values represent the means ± standard deviations (n = 3). * *p* < 0.05, ** *p* < 0.01, *** *p* < 0.001 vs. model group. AECRs: aqueous extracts of *Cibotii rhizoma*; BM: bone marrow; TPO: thrombopoietin.

**Figure 6 ijms-23-14060-f006:**
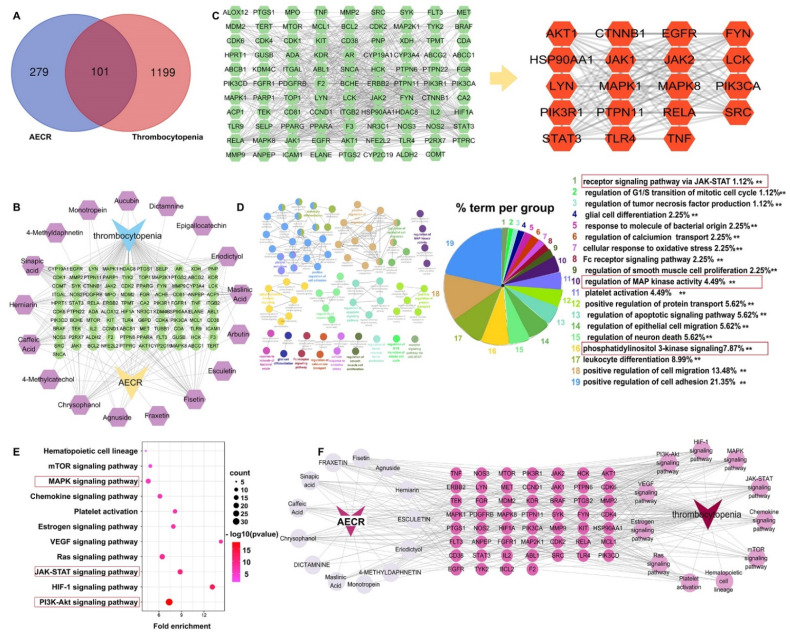
Network pharmacological prediction. (**A**) The Venn diagram of AECR targets and thrombocytopenia targets. Blue: AECR targets. Red: thrombocytopenia targets. Pink: common targets. (**B**) The AECR–ingredient–target–thrombocytopenia network. (**C**) The PPI network shows the targets of AECRs against thrombocytopenia with a combined score of >0.7, a degree of ≥23, a BC of ≥0.004028, and a CC of ≥0.443925. (**D**) GO enrichment analysis (*p* < 0.01). (**E**) Bubble plot of 11 KEGG pathways of AECRs in the treatment of thrombocytopenia. Greater enrichment is reflected in a larger fold enrichment. The size of the bubble represents the number of enriched genes in each pathway. The color of the bubble represents the Log10 range (*p*-value). (**F**) The AECR–ingredient–target–thrombocytopenia pathway network. “**” stands for *p* < 0.01. AECRs: aqueous extracts of *Cibotii rhizoma*.

**Figure 7 ijms-23-14060-f007:**
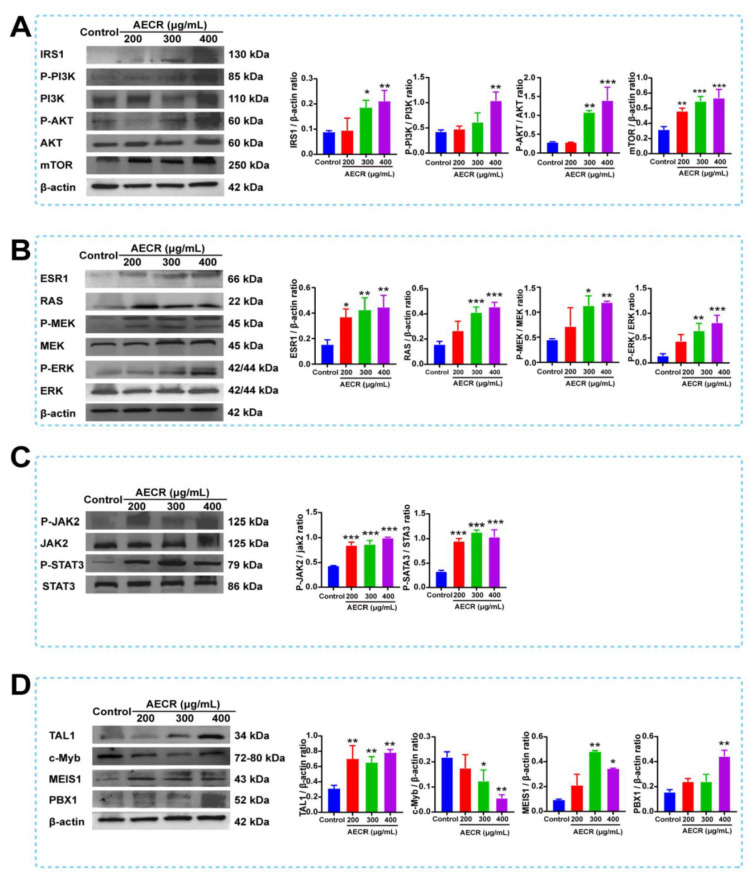
After incubating K562 cells with AECRs (200, 300, and 400 μg/mL) for 4 days, the expression of these proteins were determined by Western blot. These proteins were (**A**) IRS1, PI3K, AKT, and mTOR; (**B**) ESR1, RAS, MEK, and ERK; (**C**) JAK2 and STAT3; (**D**) TAL1, c-Myb, MEIS1, and and PBX1. On the right, this statistical histogram represents the protein level in each group. The values represent the means ± standard deviations (n = 3). * *p* < 0.05, ** *p* < 0.01, *** *p* < 0.001 versus the control group. AECRs: aqueous extracts of *Cibotii rhizoma*.

**Figure 8 ijms-23-14060-f008:**
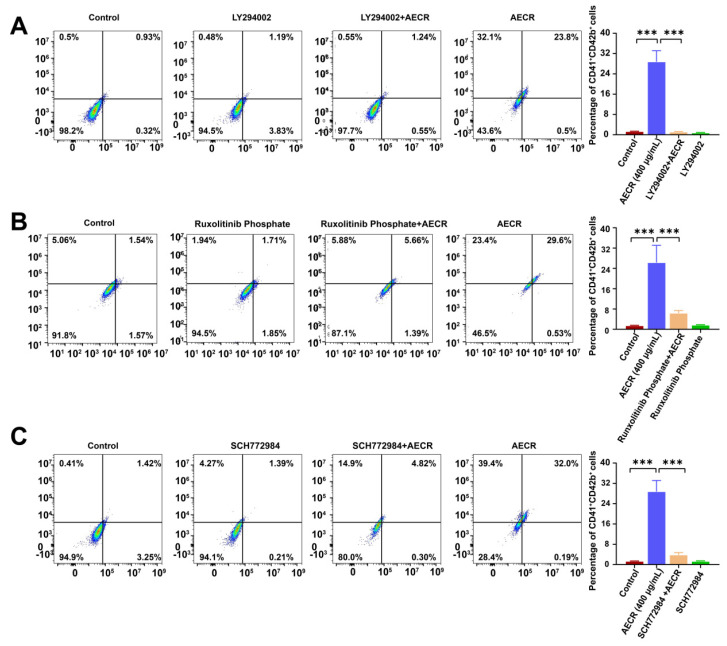
Flow cytometry was used to detect the content of CD41 and CD42b after K562 cells were incubated with inhibitors or AECRs for 4 days. The groups were set as follows: the control group, the AECR (400 μg/mL) group, the inhibitor group, and the inhibitor + AECR (400 μg/mL) group. These inhibitors were (**A**) LY294002, (**B**) ruxolitinib phosphate, and (**C**) SCH772984. On the right, this statistical histogram depicts the CD41^+^ CD42b^+^ cell proportion in each group. The values represent the means ± standard deviations (n = 3). *** *p* < 0.001 versus the control group. AECRs: aqueous extracts of *Cibotii rhizoma*.

**Figure 9 ijms-23-14060-f009:**
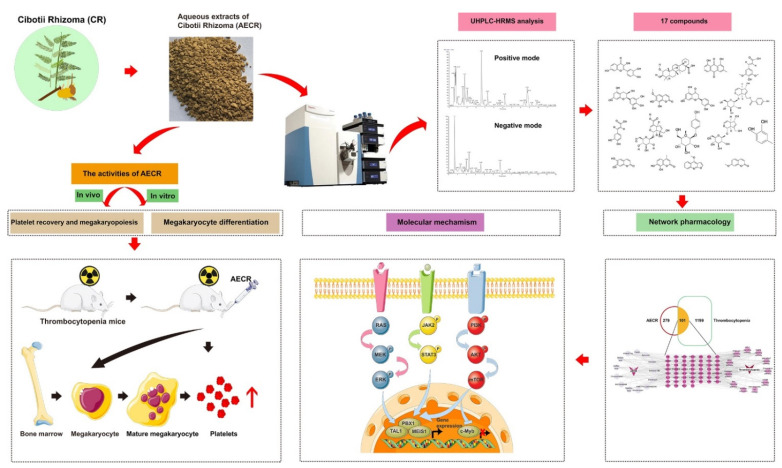
*Rhizoma cibotii (RC)* promotes megakaryopoiesis and thrombopoiesis through PI3K/AKT, MEK/ERK, and JAK2/STAT3 signaling pathways.

**Table 1 ijms-23-14060-t001:** The compounds of aqueous extracts of *Cibotii rhizoma* (AECRs).

No.	Identity	Mode	Molecular Formula	RT(min)	[M+H]^+^ (*m*/*z*)
1	Monotropein	Neg	C_16_H_22_O_11_	1.70	391.12401
2	Eriodictyol	Neg	C_15_H_12_O_6_	1.73	289.07119
3	Fraxetin	Neg	C_10_H_8_O_5_	1.89	209.04497
4	Agnuside	Neg	C_22_H_26_O_11_	1.98	467.15531
5	Sinapic acid	Neg	C_11_H_12_O_5_	2.12	225.07627
6	4-Methylcatechol	Neg	C_7_H_8_O_2_	2.30	125.06023
7	4-Methyldaphnetin	Neg	C_10_H_8_O_4_	2.53	193.05006
8	Aucubin	Neg	C_15_H_22_O_9_	2.57	347.13418
9	Dictamnine	Pos	C_12_H_9_NO_2_	2.74	200.07113
10	Esculetin	Neg	C_9_H_6_O_4_	2.76	179.03441
11	Herniarin	Pos	C_10_H_8_O_3_	2.94	177.05514
12	Arbutin	Neg	C_12_H_16_O_7_	3.09	273.0974
13	Caffeic Acid	Neg	C_9_H_8_O_4_	3.43	181.05006
14	Chrysophanol	Pos	C_15_H_10_O_4_	3.59	255.06571
15	Epigallocatechin	Neg	C_15_H_14_O_7_	4.02	307.08175
16	Fisetin	Pos	C_15_H_10_O_6_	4.28	287.05554
17	Maslinic Acid	Pos	C_30_H_48_O_4_	8.72	473.36306

## Data Availability

The data presented in this study are available in the article and Appendix A.

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
