# Peer review of "A Novel Antithrombocytopenia Agent, Rhizoma cibotii, Promotes Megakaryopoiesis and Thrombopoiesis through the PI3K/AKT, MEK/ERK, and JAK2/STAT3 Signaling Pathways"

_ijms, 2022, doi:10.3390/ijms232214060_

Round 1
Reviewer 1 Report
Dear Authors,
Thank you very much for this exciting idea, I am not sure that this idea will have a future role in the field of antithrombocytopenia to be applied to humans because in my opinion, these types of medication will take a very long time and many verifications to be approved from FDA based on previously approved medication such as Romiplostim
It will be very helpful if you can explain more about the future suggestion on this and suggest a comparison with other medications that have previous research in animals
Best regards
Author Response
Nov. 6, 2022
Dear Expert Reviewer,
Thank you very much for the prompt review process and excellent comments. We greatly appreciate the time and efforts which you have spent on it. We are submitting the revised manuscript entitled “A novel antithrombocytopenia agent, Rhizoma cibotii, promotes megakaryopoiesis and thrombopoiesis through the PI3K/AKT, MEK/ERK, and JAK2/STAT3 signaling pathways” (ID: ijms-1945106) to International Journal of Molecular Sciences.
We have carefully considered your comments and suggestions, and addressed each of the concerns in response to the comments (see point by point response). We have revised the manuscripts based on your comments and carefully checked throughout the manuscript and corrected the language errors. Our point-by-point responses to the comments (in blue) are shown below (in red).
Point. It will be very helpful if you can explain more about the future suggestion on this and suggest a comparison with other medications that have previous research in animals.
Response: Thank you very much for the excellent comments and approval of our work.
In the study, although K562 (human chronic myeloid leukemia cell) and Meg01 (human megakaryocyte) cells are considered classical models for studying megakaryocyte differentiation, they are also leukemia cells and differ to some degree from human CD34 hematopoietic progenitor cells that produce platelets. Therefore, in future work, we will construct a human CD34 hematopoietic progenitor cell model [ 1- 4] and further verify that AECR induces CD34 hematopoietic progenitor cells to produce megakaryocytes and platelets in vitro [5]. It has been reported that human bone marrow CD34 produces megakaryocytes and platelets in vitro[6]. We have added this to the discussion (page18, line 427-432).
In addition, in the revised manuscript, we have added drugs to treat thrombocytopenia (page 2-3, line 74-98). At present, the common treatment for thrombocytopenia is divided into nondrug therapy, such as platelet infusion, and drug therapy, such as thrombopoietin receptor ag-onists (TPO-Ras), recombinant human interleukin-11 (rHuIL-11) and recombinant hu-man thrombopoietin (rHuTPO). Platelet transfusion can only increase platelets in the short term, but cannot maintain platelet stability in the long term, and there is a certain risk of blood disease infection or ineffective infusion [7]. Currently, TPO-Ras, romip-lostim, eltrombopag, avatrombopag, and lusutrombopag, have been approved by the FDA and European Medicines Agency (EMA) to increase platelet counts in a variety of condi-tions, such as immune thrombocytopenia (ITP) and severe aplastic anemia [8]. They are a class of platelet growth factors that promote megakaryocyte growth, differentiation and platelet production by mimicking the effects of endogenous TPO [9]. However, long-term use can also result in adverse reactions, such as thromboembolism, bone marrow reticular fibrosis, transaminase elevation, etc. [10-11]. The drug rhTPO has shown promise in early clinical trials [12-13], but it has developed cross-reactive antibodies that neutralize native human TPO in some subjects [14-15]. Thus, the development of the drug was halted in 2001. However, recently, the potential benefits of TPO have been explored again in China [16]. Recombinant human interleukin-11 (rhIL-11, oprelvekin) received FDA approval for the prevention of Chemotherapy-induced thrombocytopenia (CIT) in patients with non-myeloid malignancies [17]. However, it has significant toxicity and high cost [17]. There-fore, searching for new drugs to treat thrombocytopenia is necessary. Many Chinese me-dicinal materials (TCMs) are derived from natural products, so it is potentially possible to find new therapeutic drugs for thrombocytopenia from natural products. And many studies have underlined natural products for the treatment of thrombocytopenia because they can be used for a long time with low cost and fewer side effects compared with chemically synthesized drugs [18].
- Choi ES, Nichol JL, Hokom MM, Hornkohl AC, Hunt P. Platelets generated in vitro from proplatelet-displaying human megakaryocytes are functional. Blood. 1995 Jan 15;85(2):402-13. PMID: 7529062.
- Bruno S, Gunetti M, Gammaitoni L, Danè A, Cavalloni G, Sanavio F, Fagioli F, Aglietta M, Piacibello W. In vitro and in vivo megakaryocyte differentiation of fresh and ex-vivo expanded cord blood cells: rapid and transient megakaryocyte reconstitution. Haematologica. 2003 Apr;88(4):379-87. PMID: 12681964.
- Iraqi M, Perdomo J, Yan F, Choi PY, Chong BH. Immune thrombocytopenia: antiplatelet autoantibodies inhibit proplatelet formation by megakaryocytes and impair platelet production in vitro. Haematologica. 2015 May;100(5):623-32. doi: 10.3324/haematol.2014.115634. Epub 2015 Feb 14. PMID: 25682608; PMCID: PMC4420211.
- Lev PR, Grodzielski M, Goette NP, Glembotsky AC, Espasandin YR, Pierdominici MS, Contrufo G, Montero VS, Ferrari L, Molinas FC, Heller PG, Marta RF. Impaired proplatelet formation in immune thrombocytopenia: a novel mechanism contributing to decreased platelet count. Br J Haematol. 2014 Jun;165(6):854-64. doi: 10.1111/bjh.12832. Epub 2014 Mar 27. PMID: 24673454.
- Perdomo J, Yan F, Leung HHL, Chong BH. Megakaryocyte Differentiation and Platelet Formation from Human Cord Blood-derived CD34+ Cells. J Vis Exp. 2017 Dec 27;(130):56420. doi: 10.3791/56420. PMID: 29364213; PMCID: PMC5908394.
- Gandhi MJ, Drachman JG, Reems JA, Thorning D, Lannutti BJ. A novel strategy for generating platelet-like fragments from megakaryocytic cell lines and human progenitor cells. Blood Cells Mol Dis. 2005 Jul-Aug;35(1):70-3. doi: 10.1016/j.bcmd.2005.04.002. PMID: 15923131.
- Sharma, S.; Sharma, P.; Tyler, L.N. Transfusion of blood and blood products: indications and complications. American family physician 2011, 83, 719-724.
- Gilreath, J.; Lo, M.; Bubalo, J. Thrombopoietin Receptor Agonists (TPO-RAs): Drug Class Considerations for Pharmacists. Drugs 2021, 81, 1285-1305, doi:10.1007/s40265-021-01553-7.
- Al-Samkari, H.; Kuter, D.J. Optimal use of thrombopoietin receptor agonists in immune thrombocytopenia. Therapeutic advances in hematology 2019, 10, 2040620719841735, doi:10.1177/2040620719841735.
- Ghanima, W.; Cooper, N.; Rodeghiero, F.; Godeau, B.; Bussel, J.B. Thrombopoietin receptor agonists: ten years later. Haematologica 2019, 104, 1112-1123, doi:10.3324/haematol.2018.212845.
- Singh, V.K.; Seed, T.M. Radiation countermeasures for hematopoietic acute radiation syndrome: growth factors, cytokines and beyond. International journal of radiation biology 2021, 97, 1526-1547, doi:10.1080/09553002.2021.1969054.
- Vadhan-Raj, S.; Verschraegen, C.F.; Bueso-Ramos, C.; Broxmeyer, H.E.; Kudelkà, A.P.; Freedman, R.S.; Edwards, C.L.; Gershenson, D.; Jones, D.; Ashby, M.; et al. Recombinant human thrombopoietin attenuates carboplatin-induced severe thrombocytopenia and the need for platelet transfusions in patients with gynecologic cancer. Annals of internal medicine 2000, 132, 364-368, doi:10.7326/0003-4819-132-5-200003070-00005.
- Moskowitz, C.H.; Hamlin, P.A.; Gabrilove, J.; Bertino, J.R.; Portlock, C.S.; Straus, D.J.; Gencarelli, A.N.; Nimer, S.D.; Zelenetz, A.D. Maintaining the dose intensity of ICE chemotherapy with a thrombopoietic agent, PEG-rHuMGDF, may confer a survival advantage in relapsed and refractory aggressive non-Hodgkin lymphoma. Annals of oncology : official journal of the European Society for Medical Oncology 2007, 18, 1842-1850, doi:10.1093/annonc/mdm341.
- Li, J.; Yang, C.; Xia, Y.; Bertino, A.; Glaspy, J.; Roberts, M.; Kuter, D.J. Thrombocytopenia caused by the development of antibodies to thrombopoietin. Blood 2001, 98, 3241-3248, doi:10.1182/blood.v98.12.3241.
- Basser, R.L.; O'Flaherty, E.; Green, M.; Edmonds, M.; Nichol, J.; Menchaca, D.M.; Cohen, B.; Begley, C.G. Development of pancytopenia with neutralizing antibodies to thrombopoietin after multicycle chemotherapy supported by megakaryocyte growth and development factor. Blood 2002, 99, 2599-2602, doi:10.1182/blood.v99.7.2599.
- Tang, B.; Huang, L.; Liu, H.; Cheng, S.; Song, K.; Zhang, X.; Yao, W.; Ning, L.; Wan, X.; Sun, G.; et al. Recombinant human thrombopoietin promotes platelet engraftment after umbilical cord blood transplantation. Blood advances 2020, 4, 3829-3839, doi:10.1182/bloodadvances.2020002257.
- Al-Samkari, H.; Soff, G.A. Clinical challenges and promising therapies for chemotherapy-induced thrombocytopenia. Expert review of hematology 2021, 14, 437-448, doi:10.1080/17474086.2021.1924053.
- Mazan, S.; Michot, B.; Bachellerie, J.P. Mouse U3-RNA-processed pseudogenes are nonrandomly integrated into genomic DNA. Implications for the process of retrogene formation. European journal of biochemistry 1989, 181, 599-605, doi:10.1111/j.1432-1033.1989.tb14766.x.
Thank you for all the valuable and helpful comments and suggestions. We hope that our revised manuscript is now suitable for publication in International Journal of Molecular Sciences.
Best regards,
Jianming Wu
Reviewer 2 Report
The research is aimed at validating the effect of traditional Chinese medicines on platelet generation and platelet count recovery in an animal model. While this study could be of high interest to the research community, the low-quality data presentation does not allow to appreciate its true value.
The manuscript has very low-quality data presentation; dozens of graphs are presented per Figure unit. It results in information overload. Moreover, all graphs are very small and as such can't be reviewed to verify the validity of the conclusions. Bar graphs are over-utilized. Please consider changing them to line charts that must be used to reflect changes in values with time.
Sample size n = 3 (for Figures 2-5, 8) is unacceptable for biomedical research where an animal model is used, and at least 6-10 repeated experiments need to be performed in order to prove the statistical significance of findings.
Many basic questions about the study methodology arise:
1) how the concentration of aqueous extract of CR was measured - based on what component it was declared to be 200-400 mg/mL.
2) Also, is it g/mL or ug/mL - these values (10^6 difference) are used on the same page - see page 4 lines 98, 99 versus page 4 lines 100, and 102. Fig 2. only depicts numbers for ug/mL concentrations. What does it mean "Safe concentration"? (page 5, line 104)
3) flow cytometry for cell culture is not described in the methods; why surface expression of receptors CD41, CD42, CD62 was measured in % of cells but not as mean fluorescence intensity, that would be more appropriate.
4) why cell viability decreases two times versus same-day control: see day 6th, 400 ug/mL for both cell lines, but apoptosis is not detected? Maybe these cells die via necrosis? or where do they go?
5) Page 114: what are big cells?
6) What features identify K562 & Meg01 differentiation and maturation? It is unclear from how results are presented and no interpretation is provided - page 6.
7) Fig 3: Why number of CD41+ and CD42+ cells increase - is it proliferation or fragmentation as a result of necrosis as suggested by a decrease of viability on days 1-6 from Figure 2A?
8) Fig. 4: Line charts must be used to depict change with time - not bar charts. What IR + AECR refers to?
9) Do not use abbreviations in figure captions.
10) Page 17 - both 10^6 different AECR concentrations are used with no explanation why it is so. For the animal model dosage is again changed - page 18.
11) Why these flow cytometric markers were chosen to analyze, please explain.
12) Conclusions are rather absent; one summary sentence does not seem sufficient to conclude the study with more than 100 charts of data.
Author Response
Nov. 6, 2022
Dear Expert Reviewer,
Thank you very much for the prompt review process and excellent comments. We greatly appreciate the time and efforts which you have spent on it. We are submitting the revised manuscript entitled “A novel antithrombocytopenia agent, Rhizoma cibotii, promotes megakaryopoiesis and thrombopoiesis through the PI3K/AKT, MEK/ERK, and JAK2/STAT3 signaling pathways” (ID: ijms-1945106) to International Journal of Molecular Sciences.
We have carefully considered your comments and suggestions, and addressed each of the concerns in response to the comments (see point by point response). We have revised the manuscripts based on your comments and carefully checked throughout the manuscript and corrected the language errors. Our point-by-point responses to the comments (in blue) are shown below (in red).
Point 1. The research is aimed at validating the effect of traditional Chinese medicines on platelet generation and platelet count recovery in an animal model. While this study could be of high interest to the research community, the low-quality data presentation does not allow to appreciate its true value.
Response 1: Thank you so much for your scientific review. We are very sorry that there are low-quality data presentation, in the revised manuscript, we changed the size of the image on (page 8, Figure 3), (page 10, Figure 4), (page 12, Figure 5), (page 14, Figure 6), and (page 18, Figure 9) to make the data and results clearer.
Point 2. The manuscript has very low-quality data presentation; dozens of graphs are presented per Figure unit. It results in information overload. Moreover, all graphs are very small and as such can't be reviewed to verify the validity of the conclusions. Bar graphs are over-utilized. Please consider changing them to line charts that must be used to reflect changes in values with time.
Response 2: Thanks a lot for the constructive and careful suggestion. We are very sorry that there are low-quality data presentation and overuse of bar charts, in the revised manuscript, we changed the size of the image on (page 8, Figure 3), (page 10, Figure 4), (page 12, Figure 5), (page 14, Figure 6), and (page 18, Figure 9) to make the data and results clearer and simpler. In addition, all the bar graphs in Figure 4B, 4C, and 4D on page 10 were changed into line graphs.
Point3. Sample size n = 3 (for Figures 2-5, 8) is unacceptable for biomedical research where an animal model is used, and at least 6-10 repeated experiments need to be performed in order to prove the statistical significance of findings.
Response 3: Thank you for your rigorous thinking. Biological research does require at least 6-10 repeated experiments to be performed in order to prove the statistical significance of findings. The animal experiment was indeed conducted according to the standard of 8 to 10 animals per group. However, since there were many indicators to be collected, and different indicators might involve the same tissues, and the number of mice in each group was limited, thus core indicators and non-core indicators were used to distinguish them. In the study, we mainly explored the therapeutic effect of drugs (AECR) on thrombocytopenia, so the statistical data of platelet was based on 8 samples in each group. And other non-core indicators, such as the expression of CD41 and CD61, and immunohistochemical, are all related to the possible mechanism in the process of platelet production, so the sample size is relatively small, with three mice per group. This approach has also been used in literature to explore the possible mechanism [1-5].
- Qu M, Fang F, Zou X, Zeng Q, Fan Z, Chen L, Yue W, Xie X, Pei X. miR-125b modulates megakaryocyte maturation by targeting the cell-cycle inhibitor p19INK4D. Cell Death Dis. 2016 Oct 20;7(10):e2430. doi: 10.1038/cddis.2016.288. PMID: 27763644; PMCID: PMC5133966.
- Wang Y, Guo Y, Zhang X, Zhao H, Zhang B, Wu Y, Zhang J. The role and mechanism of miR-557 in inhibiting the differentiation and maturation of megakaryocytes in immune thrombocytopenia. RNA Biol. 2021 Nov;18(11):1953-1968. doi: 10.1080/15476286.2021.1884783. Epub 2021 Feb 15. PMID: 33586614; PMCID: PMC8582991.
- Wang Y, Gao J, Wang H, Wang M, Wen Y, Guo J, Su P, Shi L, Zhou W, Zhou J. R-spondin2 promotes hematopoietic differentiation of human pluripotent stem cells by activating TGF beta signaling. Stem Cell Res Ther. 2019 May 20;10(1):136. doi: 10.1186/s13287-019-1242-9. PMID: 31109354; PMCID: PMC6528258.
- Juban G, Sakakini N, Chagraoui H, Cruz Hernandez D, Cheng Q, Soady K, Stoilova B, Garnett C, Waithe D, Otto G, Doondeea J, Usukhbayar B, Karkoulia E, Alexiou M, Strouboulis J, Morrissey E, Roberts I, Porcher C, Vyas P. Oncogenic Gata1 causes stage-specific megakaryocyte differentiation delay. Haematologica. 2021 Apr 1;106(4):1106-1119. doi: 10.3324/haematol.2019.244541. PMID: 32527952; PMCID: PMC8018159.
- Weiss CN, Ito K. microRNA-22 promotes megakaryocyte differentiation through repression of its target, GFI1. Blood Adv. 2019 Jan 8;3(1):33-46. doi: 10.1182/bloodadvances.2018023804. PMID: 30617215; PMCID: PMC6325298.
Point 4. Many basic questions about the study methodology arise:
1) how the concentration of aqueous extract of CR was measured - based on what component it was declared to be 200-400 mg/mL.
Response 4.1: Thanks for your careful review. The concentration of 200 - 400 μg/ml was obtained by weighing an appropriate amount of the drug (AECR) and dissolved in sterile deionized water. In brief, (AECR mass) / (water volume). Meanwhile, we have added the AECR preparation and administration description in the materials and methods section (page 18, lines 436-439).
2) Also, is it g/mL or ug/mL - these values (10^6 difference) are used on the same page - see page 4 lines 98, 99 versus page 4 lines 100, and 102. Fig 2. only depicts numbers for ug/mL concentrations. What does it mean "Safe concentration"? (page 5, line 104)
Response 4.2: Thank you for your reminding. We are very sorry for the omission or miswriting of the concentration unit on the same page and we have uniformly modified the concentration unit to be μg/mL in the revised version (page 5, line 127-135).
For the “safe concentration”, in this paper, it mainly means that these concentrations do not damage the cells. It is elaborated through Cell Counting Kit-8 (CCK-8) and Lactate dehydrogenase cytotoxicity (LDH) methods. CCK-8 method: the dehydrogenase in living cells is used to catalyze WST-8 reagent to produce dirty dye. The amount of dirty dye is in a linear relationship with the number of living cells, but the dead cells can't catalyze WST-8 reagent to produce dirty dye. Therefore, with the increase in the concentration of AECR (200, 300, and 400 μg/mL), the number of cells was inhibited, and the percentage of cell viability decreased.
LDH method: cell membrane structure damage caused by apoptosis or necrosis can result in the release of stable LDH in cell cytoplasm into the culture medium. Quantitative analysis of cytotoxicity can then be achieved by detecting the activity of LDH released into the culture medium. In this experiment, there were no significant changes in the release amount of LDH in the AECR (200, 300, 400 μg/mL) groups compared with the control group, respectively, so these three concentrations are non-toxic or safe.
3) flow cytometry for cell culture is not described in the methods; why surface expression of receptors CD41, CD42, CD62 was measured in % of cells but not as mean fluorescence intensity, that would be more appropriate.
Response 4.3: We gratefully appreciate your valuable suggestion. We have added this method to the analysis of cell surface antigen expression by flow cytometry (page 20, lines 502-511). The expression of cell surface antigen was analyzed by flow cytometry as follows: cells (3×104 cells/mL) were treated with or without AECR (200, 300, and 400 µg/mL) for 4 days. The cells were then harvested, and resuspended in 100 μL of ice-cold PBS and followed by the addition of 3 μL FITC-CD41 and PE-CD42b antibodies (Biolegend, San Diego, CA, USA), respectively, and incubated for 30 min at room temperature in the dark. Thereafter, 1 × 105 cells were resuspended in 500 μL of ice-cold PBS for flow cytometry analysis (BD Biosciences, CA).
In this study, we aim to induce the differentiation of K562 and Meg01 cells into megakaryocytes through AECR. And the effect of the drug is linear with the number of megakaryocytes, so we use the percentage of cells. In addition to, our default parameters were unified as total number of cells: 10000 cells, next the number of CD41+, CD42b+, or CD62P+ cells were analyzed. And there have been reports of similar statistical methods [1-2].
- Mazzi S, Dessen P, Vieira M, Dufour V, Cambot M, El Khoury M, Antony-Debré I, Arkoun B, Basso-Valentina F, BenAbdoulahab S, Edmond V, Rameau P, Petermann R, Wittner M, Cassinat B, Plo I, Debili N, Raslova H, Vainchenker W. Dual role of EZH2 in megakaryocyte differentiation. Blood. 2021 Oct 28;138(17):1603-1614. doi: 10.1182/blood.2019004638. PMID: 34115825; PMCID: PMC8554649.
- Valet C, Magnen M, Qiu L, Cleary SJ, Wang KM, Ranucci S, Grockowiak E, Boudra R, Conrad C, Seo Y, Calabrese DR, Greenland JR, Leavitt AD, Passegué E, Méndez-Ferrer S, Swirski FK, Looney MR. Sepsis promotes splenic production of a protective platelet pool with high CD40 ligand expression. J Clin Invest. 2022 Apr 1;132(7):e153920. doi: 10.1172/JCI153920. PMID: 35192546; PMCID: PMC8970674.
4) why cell viability decreases two times versus same-day control: see day 6th, 400 ug/mL for both cell lines, but apoptosis is not detected? Maybe these cells die via necrosis? or where do they go?
Response 4.4: Thank you for your rigorous thinking. AECR (400 μg/mL) decreased cell viability, which may be due to the inhibitory effect of high concentration on cell proliferation. And it has been reported that during the differentiation of megakaryonuclear progenitor cells, the cells undergo endomitosis, leading to the increase of polyploid megakaryocyte cells without increasing the number of cells [1]. The following diagram shows mitosis leading to proliferation and endomitosis leading to polyploid cells.
|
The figure please find at the attachment file. |
- Mazzi S, Lordier L, Debili N, Raslova H, Vainchenker W. Megakaryocyte and polyploidization. Exp Hematol. 2018 Jan;57:1-13. doi: 10.1016/j.exphem.2017.10.001. Epub 2017 Oct 27. PMID: 29111429.
5) Page 114: what are big cells?
Response 4.5: We are sorry about the use of "big cell" in the article, because it is mainly used as a colloquial term. In the revised manuscript, we have changed the colloquial term "big cell" to the written term "polyploid cell" (page 6-7, lines 147-151).
6) What features identify K562 & Meg01 differentiation and maturation? It is unclear from how results are presented and no interpretation is provided - page 6.
Response 4.6: Thank you for your rigorous consideration. K562 and Meg01 cells are classical models for exploring megakaryocytic differentiation. Megakaryocyte differentiation is characterized by increasing cell size and cell polyploidy, the expression of cell surface specific markers, including CD41 and CD42b, etc. [1-3]. We added the characteristics of megakaryocytes in differentiation and maturation and elaborated on the significance of each picture to make the results clearer (page 6-7, line 146-164).
|
The figure please find at the attachment file. |
|
The figure please find at the attachment file. |
|
The figure please find at the attachment file. |
|
The figure please find at the attachment file. |
- Deutsch VR, Tomer A. Megakaryocyte development and platelet production. Br J Haematol. 2006 Sep;134(5):453-66. doi: 10.1111/j.1365-2141.2006.06215.x. PMID: 16856888.
- Yen JH, Lin CY, Chuang CH, Chin HK, Wu MJ, Chen PY. Nobiletin Promotes Megakaryocytic Differentiation through the MAPK/ERK-Dependent EGR1 Expression and Exerts Anti-Leukemic Effects in Human Chronic Myeloid Leukemia (CML) K562 Cells. Cells. 2020 Apr 3;9(4):877. doi: 10.3390/cells9040877. PMID: 32260160; PMCID: PMC7226785.
- Noetzli LJ, French SL, Machlus KR. New Insights Into the Differentiation of Megakaryocytes From Hematopoietic Progenitors. Arterioscler Thromb Vasc Biol. 2019 Jul;39(7):1288-1300. doi: 10.1161/ATVBAHA.119.312129. Epub 2019 May 2. PMID: 31043076; PMCID: PMC6594866.
7) Fig 3: Why number of CD41+ and CD42+ cells increase - is it proliferation or fragmentation as a result of necrosis as suggested by a decrease of viability on days 1-6 from Figure 2A?
|
The figure please find at the attachment file. |
|
The figure please find at the attachment file. |
Response 4.7: Thanks for your careful review. CD41 and CD42b are surface markers of megakaryocytes (Response 4.6 FigureB). The increase of CD41+ and CD42b+ cells indicated that AECR promoted the differentiation and maturation of K562 and Meg01 egakaryocyte progenitor cells into megakaryocytes. For the reasons for the decline in cell viability shown in Figure 2A, see response 4.2 and response 4.4 above. CD41 and CD42b flow scatter plots were also provided to confirm that the increase in the number of CD41+ and CD42b+ cells was not a false positive due to cell debris.
8) Fig. 4: Line charts must be used to depict change with time - not bar charts. What IR + AECR refers to?
Response 4.8: Thank you very much for your advice. In the revised manuscript, the bar graphs in Figure 4B, 4C, and 4D (page10, lines 206-207 ) were changed into line graphs. In addition to, “IR + AECR” refers to the treatment group of AECR for radiation-induced thrombocytopenia, where “IR” stands for ionizing radiation and “AECR” stands for the drug. In the revised manuscript, IR+AECR is supplemented and explained (page 20, lines 524-534).
9) Do not use abbreviations in figure captions.
Response 4.9: Thank you so much for your careful check. In the revised version, we have changed the abbreviations in figure captions (Page 6, Figure2), (Page 9, Figure3), (Page 10, Figure4), (Page 13, Figure5), (Page 14, Figure6), (Page 16, Figure7 and Figure8).
10) Page 17 - both 10^6 different AECR concentrations are used with no explanation why it is so. For the animal model dosage is again changed - page 18.
Response 4.10: At the preliminary screening stage, we used a widely range of drug concentrations to identify the activities of AECR in vitro and in vivo. Pre-experiment results showed that 200, 300 and 400 µg/mL AECR had positive effects on megakaryocyte differentiation with no cytotoxicity, and 143, 286 and 429 mg/kg AECR had therapeutic effects on radiation-induced thrombocytopenia mice without system toxicity as manuscript described. The lower concentrations of AECR had no effects on megakaryocyte differentiation and platelet production, while higher concentrations of AECR could produce toxic effects in vitro and in vivo.
11) Why these flow cytometric markers were chosen to analyze, please explain.
Response 4.11: In this study, the cell surface antigens CD41, CD42b, CD61, CD62P, and c-Kit (CD117) were detected by flow cytometry because they play an important role in megakaryocyte differentiation and platelet production [1-6]. During this process, the antigen on cell surface changes as shown in the figure below. That is, by detecting the expression of these antigens, we can speculate about the mechanism of drug AECR therapy for radiation-induced thrombocytopenia.
- Ferkowicz MJ, Starr M, Xie X, Li W, Johnson SA, Shelley WC, Morrison PR, Yoder MC. CD41 expression defines the onset of primitive and definitive hematopoiesis in the murine embryo. Development. 2003 Sep;130(18):4393-403. doi: 10.1242/dev.00632. PMID: 12900455.
- Rhodes KE, Gekas C, Wang Y, Lux CT, Francis CS, Chan DN, Conway S, Orkin SH, Yoder MC, Mikkola HK. The emergence of hematopoietic stem cells is initiated in the placental vasculature in the absence of circulation. Cell Stem Cell. 2008 Mar 6;2(3):252-63. doi: 10.1016/j.stem.2008.01.001. PMID: 18371450; PMCID: PMC2888040.
- Robin C, Ottersbach K, Boisset JC, Oziemlak A, Dzierzak E. CD41 is developmentally regulated and differentially expressed on mouse hematopoietic stem cells. Blood. 2011 May 12;117(19):5088-91. doi: 10.1182/blood-2011-01-329516. Epub 2011 Mar 17. PMID: 21415271; PMCID: PMC3109535.
- Gekas C, Graf T. CD41 expression marks myeloid-biased adult hematopoietic stem cells and increases with age. Blood. 2013 May 30;121(22):4463-72. doi: 10.1182/blood-2012-09-457929. Epub 2013 Apr 5. PMID: 23564910.
- Lin J, Zeng J, Liu S, Shen X, Jiang N, Wu YS, Li H, Wang L, Wu JM. DMAG, a novel countermeasure for the treatment of thrombocytopenia. Mol Med. 2021 Nov 27;27(1):149. doi: 10.1186/s10020-021-00404-1. PMID: 34837956; PMCID: PMC8626956.
- Yang X, Chitalia SV, Matsuura S, Ravid K. Integrins and their role in megakaryocyte development and function. Exp Hematol. 2022 Feb;106:31-39. doi: 10.1016/j.exphem.2021.11.007. Epub 2021 Dec 12. PMID: 34910941; PMCID: PMC8795491.
12) Conclusions are rather absent; one summary sentence does not seem sufficient to conclude the study with more than 100 charts of data.
Response 4.12: Thanks for your careful review. In the revised draft, we have enriched and improved the conclusions (page22-23, lines 627-637). Conclusion: In this study, we obtained for the first time 17 possible components of AECR by UHPLCHRMS. Secondly, network pharmacology predicted that the molecular mechanism of AECR in the treatment of thrombocytopenia was closely related to PI3K/AKT, JAK/STAT, RAS, MAPK and mTOR pathways. In vitro experiments confirmed that AECR promotes megakaryocyte differentiation by activating PI3K/AKT, MEK/ERK, and JAK2/STAT3 signaling pathways. In vivo experiments confirmed that AECR promotes differentiation and maturation of megakaryocytes and platelet production. Finally, it is concluded that AECR may treat radiation-induced thrombocytopenia by promoting megakaryocyte differentiation to produce platelets. These indicate that it may be a new drug for the treatment of thrombocytopenia and lay a foundation for future clinical application.
Thank you for all the valuable and helpful comments and suggestions. We hope that our revised manuscript is now suitable for publication in International Journal of Molecular Sciences.
Best regards,
Jianming Wu
Round 2
Reviewer 1 Report
Dear Authors,
Thank you for considering my comments and revising the needed details as well as providing me with the required explanation, I hope this trial will open the door for another research and new ways in this regard
Best of luck
Author Response
Thank you very much for the encouragement and approval of our work.
Reviewer 2 Report
The authors have carefully reviewed and responded to the suggestions of the Round 1 review. The improved version of the manuscript presentation is of much better quality and the clarity of the figures has been significantly approved. The sample size remains a concern, however. While the scientific reasoning and logistics behind it are understandable, I suggest disclosing the small sample size (n=3) and reasoning behind it as a limitation of the study.
Author Response
Dear Expert Reviewer,
Thank you very much for the prompt review process and excellent comments. We greatly appreciate the time and efforts which you have spent on it. We are submitting the revised manuscript entitled “A novel antithrombocytopenia agent, Rhizoma cibotii, promotes megakaryopoiesis and thrombopoiesis through the PI3K/AKT, MEK/ERK, and JAK2/STAT3 signaling pathways” (ID: ijms-1945106) to International Journal of Molecular Sciences.
We have carefully considered your comments and suggestions, and addressed each of the concerns in response to the comments (see point by point response). We have revised the manuscripts based on your comments and carefully checked throughout the manuscript and corrected the language errors. Our point-by-point responses to the comments (in blue) are shown below (in red).
Point2. The authors have carefully reviewed and responded to the suggestions of the Round 1 review. The improved version of the manuscript presentation is of much better and the clarity of the figures has been significantly approved. The sample size remains a concern, however. While the scientific reasoning and logistics behind it are understandable, I suggest disclosing the small sample size (n=3) and reasoning behind it as a limitation of the study.
Response 1: Thanks a lot for the constructive and careful suggestion. We clearly noted the annotations of the small sample size (page 6, lines 138-142), (page 9, lines 183-184) (page 13, lines 266-278) and discussed the reasons for this (page 18, lines 437-457). The details are as follows: in summary, although the current data suggest that AECR may treat radiation-induced thrombocytopenia by promoting megakaryocyte differentiation and maturation, the argument for this conclusion is still worthy of consideration and further confirmation. In the study, although K562 (human chronic myeloid leukemia cell) and Meg01 (human megakaryocyte) cells are considered classical models for studying megakaryocyte differentiation, they are also leukemia cells and differ to some degree from human CD34 hematopoietic progenitor cells that produce platelets. Human bone marrow CD34 hematopoietic stem cells can be induced to produce platelets in vitro[1]. Therefore, the demonstration of the mechanism in vitro still requires a large number of biological replicates, as well as the construction of other cell models[2-5], such as human bone marrow CD34 hematopoietic stem cells, for demonstration and study in the future. In vivo, our data suggest that AECR can treat radiation-induced thrombocytopenia by promoting hematopoietic stem progenitor cells to mature into megakaryocytes. This conclusion is significant because TPO-RAs drugs, romiplostim, eltrombopag, avatrombopag, and lusutrombopag, are currently approved by the FDA and EMA for the treatment of thrombocytopenia, which their mechanism was by promoting megakaryocyte growth, differentiation and platelet generation[6-7]. However, there are some limitations in the conclusions of this paper, such as the small sample size (n=3) and shallow mechanism studies. Therefore, to further verify and deepen the mechanism by which AECR promotes megakaryocyte differentiation to produce platelets, a large number of biological repeats and other methods[8] are needed in the future to deepen the mechanism research.
- Perdomo J, Yan F, Leung HHL, Chong BH. Megakaryocyte Differentiation and Platelet Formation from Human Cord Blood-derived CD34+ Cells. J Vis Exp. 2017 Dec 27;(130):56420. doi: 10.3791/56420. PMID: 29364213; PMCID: PMC5908394.
- Choi ES, Nichol JL, Hokom MM, Hornkohl AC, Hunt P. Platelets generated in vitro from proplatelet-displaying human megakaryocytes are functional. Blood. 1995 Jan 15;85(2):402-13. PMID: 7529062.
- Bruno S, Gunetti M, Gammaitoni L, Danè A, Cavalloni G, Sanavio F, Fagioli F, Aglietta M, Piacibello W. In vitro and in vivo megakaryocyte differentiation of fresh and ex-vivo expanded cord blood cells: rapid and transient megakaryocyte reconstitution. Haematologica. 2003 Apr;88(4):379-87. PMID: 12681964.
- Iraqi M, Perdomo J, Yan F, Choi PY, Chong BH. Immune thrombocytopenia: antiplatelet autoantibodies inhibit proplatelet formation by megakaryocytes and impair platelet production in vitro. Haematologica. 2015 May;100(5):623-32. doi: 10.3324/haematol.2014.115634. Epub 2015 Feb 14. PMID: 25682608; PMCID: PMC4420211.
- Lev PR, Grodzielski M, Goette NP, Glembotsky AC, Espasandin YR, Pierdominici MS, Contrufo G, Montero VS, Ferrari L, Molinas FC, Heller PG, Marta RF. Impaired proplatelet formation in immune thrombocytopenia: a novel mechanism contributing to decreased platelet count. Br J Haematol. 2014 Jun;165(6):854-64. doi: 10.1111/bjh.12832. Epub 2014 Mar 27. PMID: 24673454.
- Gilreath, J.; Lo, M.; Bubalo, J. Thrombopoietin Receptor Agonists (TPO-RAs): Drug Class Considerations for Pharmacists. Drugs 2021, 81, 1285-1305, doi:10.1007/s40265-021-01553-7.
- Al-Samkari, H.; Kuter, D.J. Optimal use of thrombopoietin receptor agonists in immune thrombocytopenia. Therapeutic advances in hematology 2019, 10, 2040620719841735, doi:10.1177/2040620719841735.
- Salzmann M, Hoesel B, Haase M, Mussbacher M, Schrottmaier WC, Kral-Pointner JB, Finsterbusch M, Mazharian A, Assinger A, Schmid JA. A novel method for automated assessment of megakaryocyte differentiation and proplatelet formation. Platelets. 2018 Jun;29(4):357-364. doi: 10.1080/09537104.2018.1430359. Epub 2018 Feb 20. PMID: 29461915.
Thank you for all the valuable and helpful comments and suggestions. We hope that our revised manuscript is now suitable for publication in International Journal of Molecular Sciences.
Best regards,
Jianming Wu